# Amazonian understory forests change phosphorus acquisition strategies under elevated CO$_2$

The potential for the Amazon forest to continue functioning as a carbon (C) sink strongly depends on soil nutrient availability, particularly phosphorus (P), and on plants' ability to adjust nutrient acquisition strategies. However, limited experimental evidence constrains the mechanistic representation of nutrient–carbon interactions in climate change models. Here, we conducted an experiment in a P-depleted Amazonian understory forest, increasing atmospheric CO$_2$ in-situ by approximately 300 ppm using open-top chambers. We show that elevated CO$_2$ (eCO$_2$) induced contrasting responses by roots along the litter–soil continuum that could facilitate nutrient uptake. Litter-layer roots maintained net productivity but increased specific root length under eCO$_2$, indicating enhanced foraging efficiency. In contrast, soil-based roots reduced productivity but showed increased arbuscular mycorrhizal colonization. Additionally, eCO$_2$ caused a significant decline in soil organic P. Our findings suggest that eCO$_2$ intensifies competition for P between plant roots and soil microorganisms, leading to changes in litter and soil P pools and exacerbating already strong nutrient constraints. Such spatially divergent adjustments in root nutrient acquisition strategies may critically regulate plant soil C and P coupling and should be incorporated into assessments of Amazon forest resilience under future climate change.

The Amazon forest plays a crucial role as a terrestrial C sink and has a high potential to mitigate the effects of rising atmospheric CO$_2$ levels[1]. Earth system models project an increased C sink of tropical vegetation for the next decades[2], based on an expected stimulation of photosynthesis by eCO$_2$, referred to as the 'CO$_2$ fertilization effect'[3]. However, vegetation models with implemented nutrient feedback[4], and experimental data from free-air-CO$_2$-enrichment (FACE) experiments in several forest ecosystems suggest that the magnitude of a potential CO$_2$ fertilization effect depends on plant adaptations, soil nutrient status[5–7] and plant-microbe interactions[8,9]. Approximately 60% of Amazonian forests grow on geologically old, P-depleted soils[10], where low P availability limits net productivity[11,12]. However, despite these P constraints, understory plants in Central Amazonia substantially increased growth under eCO$_2$, enhancing C assimilation by 67%, leaf area by 51%, and stem diameter by 65%, indicating a 'CO$_2$ fertilization

effect' on aboveground growth[13]. The mechanisms driving this CO$_2$ fertilization effect remain unclear, but they are likely enabled by adaptations at the plant–soil interface. A clear understanding is crucial for improving nutrient feedback in vegetation models and predicting tropical forest resilience under climate change[4].

In many highly weathered soils of the Amazon basin, a large fraction of P is not accessible and strongly adsorbed to the soil matrix (residual P), adsorbed to iron and aluminum oxides (mineral P), or bound in organic forms (Po), and only a relatively small inorganic P (Pi) fraction in the soil solution is accessible to plants and microbial communities[10,14]. Forests growing on such soils have a unique high taxonomic and functional plant biodiversity[15], and a similarly high root functional diversity[16]. In soils with rather low P concentrations, plants likely prioritize P mining strategies via phosphatases and organic acid exudation, whereas in substrates with intermediate to high P

✉ e-mail: nathielly.martins@tum.de; lucia.fuchslueger@univie.ac.at

concentrations, investments are primarily directed to P foraging strategies via root morphological adaptations and in collaboration with arbuscular mycorrhizas (AMF)[17]. A phenomenon particularly pronounced in Amazonian forests is high root proliferation within the litter layer[18,19], which contains significant amounts of Po[20]. Litter-based roots have around 15% higher specific root length (SRL) and a 32% higher root tip abundance compared to soil-based roots[21]. In addition, they have been found to stimulate the mineralization of Po from decaying leaves and wood through phosphatase enzymes[19,22]. This trait combination can aid plants in assimilating newly hydrolyzed Pi from the litter layer before it becomes bound to the soil matrix[18,19,22]. Importantly, plants can adjust their root systems in response to changing resource availability[21,23]. On a plant community level, combinations of root nutrient acquisition strategies in a multi-dimensional 'root economics space' can inform about predominant nutrient acquisition strategies[24,25] and link root (morphological) form to function, define C-cost investments and trait trade-offs[26]. Hence, changes at the root level, resulting in shifts in root trait combinations, infer cascading effects on C inputs to soil and root nutrient uptake.

Carbon allocated to soil via roots provides substrate for soil microbial communities; in particular, labile exudates can enhance organic matter decomposition in the rhizosphere (i.e., rhizosphere priming effect)[27] by providing extra energy for producing extracellular enzymes by microbes (for instance, glucosidases, cellobiosidases, or glucosaminidases) to accelerate the hydrolysis of higher polymeric and more complex organic matter compounds[28]. Similarly, microbes can release phosphatases to mineralize Po and immobilize Pi in their biomass, which can, in low P ecosystems, significantly contribute to the total soil P budget[29]. On the other hand, C released into the soil, assimilated by microbial cells, can enter the microbial necromass loop or form soil organic matter[30]. In this way, interactions at the root-

microbe interface control soil organic C dynamics and could be key to longer-term sequestration in a future climate[31].

Here, we investigate how rising atmospheric $CO_2$ affects plant-soil-microbe interactions and P acquisition of an understory plant community adapted to low-P soils. In contrast to a greenhouse experiment with different tropical species[32], and a FACE experiment conducted in a low P, monodominant mature *Eucalyptus* subtropical forest[7], we previously found that $eCO_2$ not only increased C assimilation, but also above-ground biomass growth in native tropical understory forest trees[13]. Hence, the overarching goal of this work is to unravel the mechanisms and adaptations at the root and soil interface that facilitated this stark $CO_2$ fertilization response. We hypothesize that under $eCO_2$, plants have used the extra C available to i) increase soil root biomass and productivity, ii) further invest in mining P from Po-rich sources like the organic litter layer, and iii) increase investment in AMF symbioses. In addition, we investigate the effects of $CO_2$ fertilization on soil nutrient composition and microbial community dynamics to gain a more comprehensive understanding of $eCO_2$. The $CO_2$ enrichment experiment is conducted with native, mature, tropical lowland forest understory tree communities growing on P-depleted soils as a model system using Open-Top Chambers (OTC) at the study site of the AmazonFACE program, located approximately 70 km north of Manaus, Brazil[33]. A total of eight OTCs (with a 2.4 m diameter and 3 m in height) were established in four pairs, distributed across a plateau within a primary forest. This results in four OTCs with ambient $CO_2$ (a$CO_2$) and four with elevated $CO_2$ by ~300 ppm ($eCO_2$) relative to the paired a$CO_2$ OTC. Compared to canopy or emergent trees, tropical understory trees usually grow more slowly as an adaptation to light limitation[34]. Nevertheless, understory trees contribute approximately 20% to the total tropical forest C sink[35] and play a key role in forest responses to climate change[36], emphasizing the need for a deeper understanding of their functionality and dynamics.

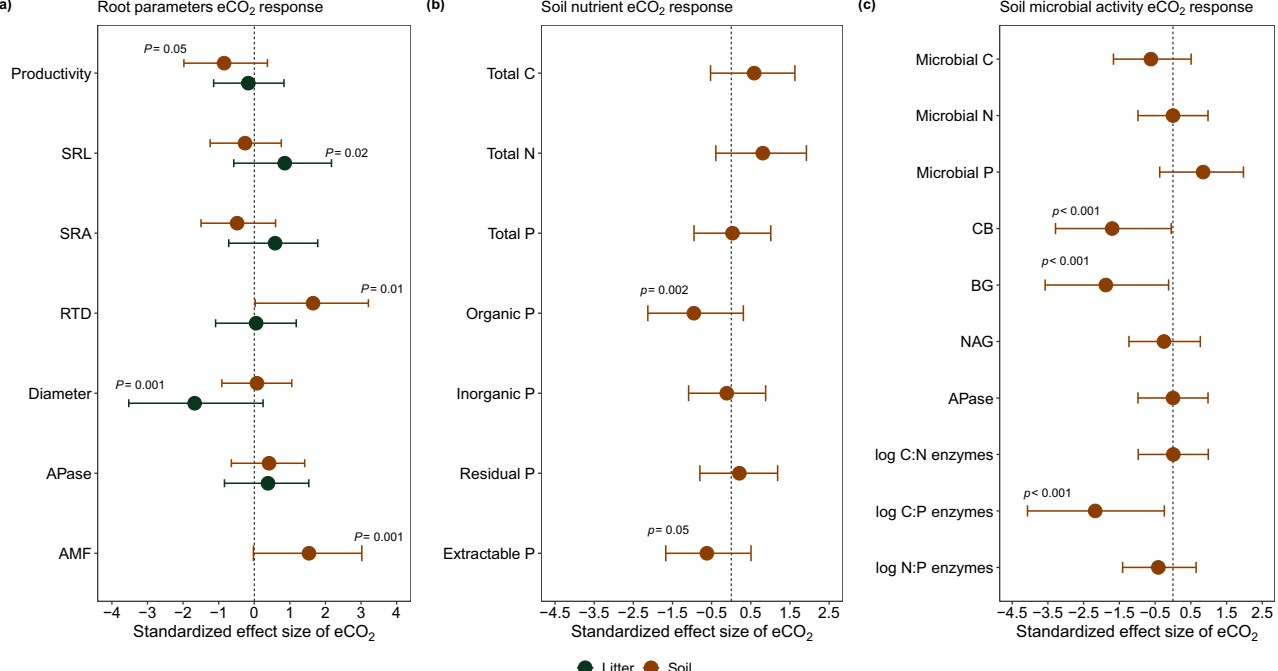

**Fig. 1 | Elevated $CO_2$ effects on root parameters, soil nutrient concentration, and microbial activity.** (a) fine root parameters, (b) soil nutrient concentration, and (c) soil microbial activity. The effects of $eCO_2$ are represented as the 'delta $CO_2$ effect' (i.e., ambient $CO_2$ treatment (a$CO_2$)−ambient $CO_2$ baseline, and $eCO_2$ treatment−$eCO_2$ baseline; see "methods" section for a detailed description) and shown as a standardized effect size of $eCO_2$ compared with the respective a$CO_2$. The circles indicate changes of stocks or processes in the litter layer (dark green) and the soil (saddle brown) and show the average response to $eCO_2$ ( $n$ = 4); circles on the left and lower than zero indicate a negative effect of $eCO_2$, circles on the

right a positive effect of $eCO_2$, the error bars represent the 95% confidence interval (CI). SRL specific root length, SRA specific root area, RTD root tissue density, Root APase acid phosphomonoesterase activity. Soil nutrient concentrations are reported as C (carbon), N (nitrogen), and P (phosphorus) and different P soil fractions (organic, inorganic, residual, and extractable). Changes in soil microbial activity rates are reported as CNP microbial biomass and the microbial extracellular enzymes: CB cellobiosidase; BG β-glucosidase; NAG Exochitinase ($N$-acetyl-β-glu-cosaminidase). The $P$ values obtained by linear generalized mixed models are indicated in the variables where the $eCO_2$ significantly differed from the a$CO_2$.

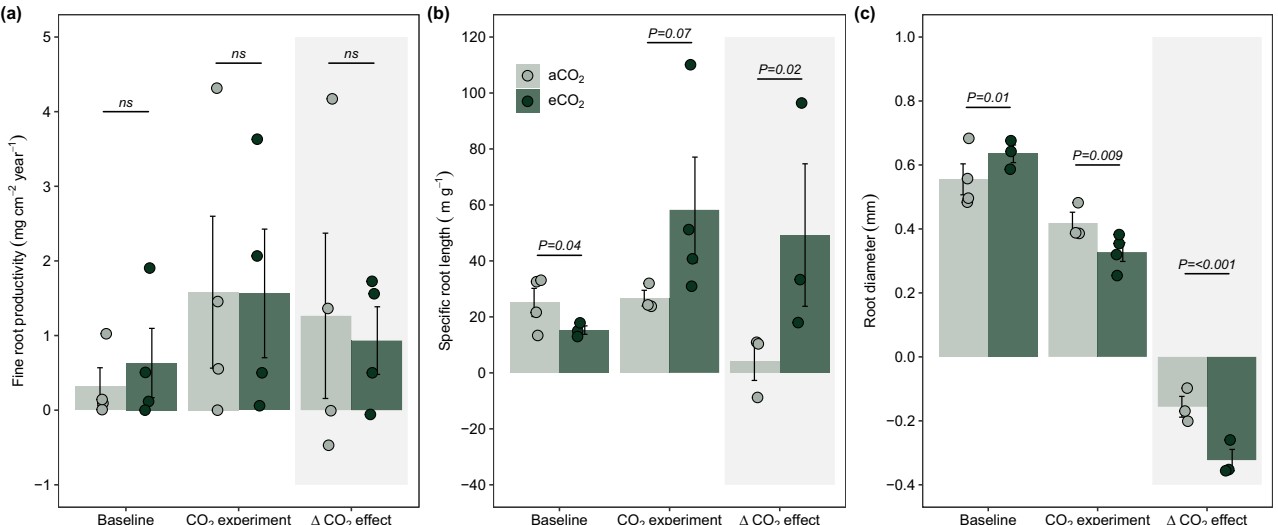

**Fig. 2 | Root dynamics in the litter layer under elevated CO₂. (a)** fine root productivity, **(b)** specific root length (SRL), and **(c)** root diameter. We present the baseline collections conducted before the start of elevated CO₂, the collection after elevated CO₂ was initiated (i.e., CO₂ experiment; see more details in Fig. S2), and the delta CO₂ effect. The 'delta CO₂ effect' was calculated to account for effects caused by spatial and temporal variability as the difference between the "CO₂ experiment" and the "baseline" (see "Methods" section). Data were analyzed using generalized linear mixed models, with a specific model developed for each variable. The treatment was considered a fixed factor, and spatial variability was controlled by including the pair of open-top chambers (OTC) as a random factor. The numbers at the top of the bars indicate the probability of no difference between the means for each comparison. The bars represent the mean and standard error of $n = 4$, and the points indicate the samples for each group.

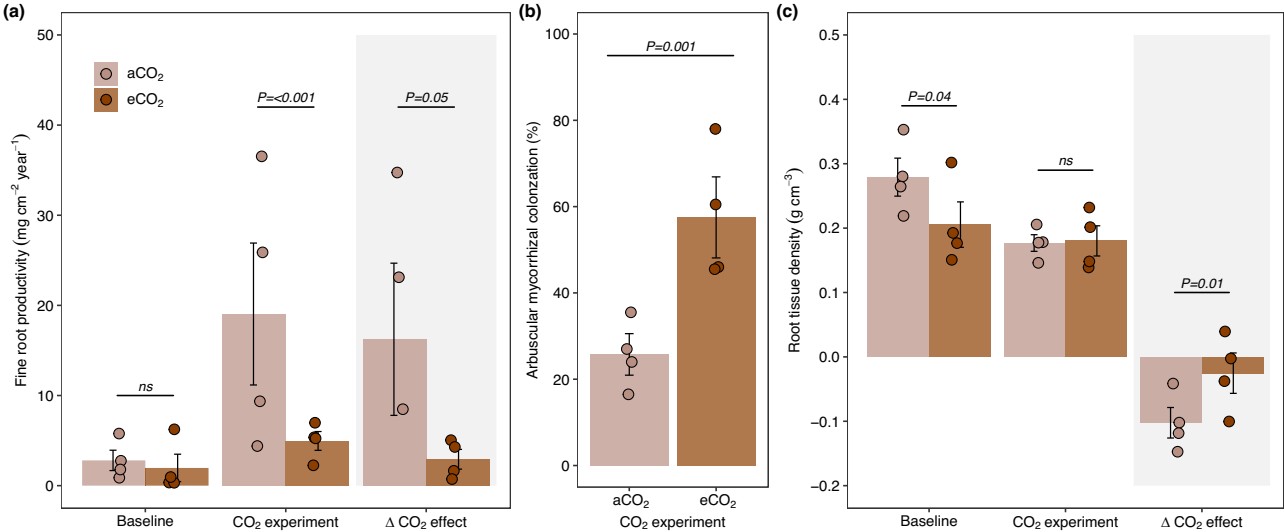

**Fig. 3 | Root dynamics up to 15 cm in the soil layer under elevated CO₂. (a)** fine root productivity, **(b)** arbuscular mycorrhizal colonization (AMF), and **(c)** root tissue density (RTD). We present the baseline collections conducted before the start of elevated CO₂, the collection after elevated CO₂ was initiated (i.e., CO₂ experiment; see more details in Fig. S2), and the delta CO₂ effect. The 'delta CO₂ effect' was calculated to account for effects caused by spatial and temporal variability as the difference between the "CO₂ experiment" and the "baseline" (see "Methods" section). Data were analyzed using generalized linear mixed models, with a specific model developed for each variable. The treatment was considered a fixed factor, and spatial variability was controlled by including the pair of open-top chambers (OTC) as a random factor. The numbers at the top of the bars indicate the probability of no difference between the means for each comparison. The bars represent the mean and standard error of $n = 4$, and the points indicate the samples for each group.

## Results and discussion

### Root trait adaptations for efficient nutrient acquisition in response to eCO₂

The studied understory plant communities adjusted root P acquisition strategies in response to eCO₂ differentially in the litter and soil layer. Although eCO₂ did not change the net productivity of litter layer-based roots after eight months (Figs. 1a, 2a), we observed a change in fine root morphological traits. More precisely, we found that eCO₂ increased SRL (Figs. 1a, 2b; $Z = 2.22$, $p = 0.02$) and decreased root diameter significantly

(Fig. 1a, 2c; $Z = -4.64$, $p < 0.001$). As root tips and root apices are the hotspots of root exudation and nutrient uptake[37], by producing long and fine roots, plants can, in this way, increase both substrate-root interaction surfaces and the volume of substrate intercepted[37,38] and facilitate the interception of scarce nutrients during foraging[19]. Root APase, expressed per dry root mass, root length, and root area, did not change with eCO₂ in litter-based roots (Fig. S6a–c). However, it has been shown that long and fine roots are associated with high phosphatase activity[39]. This means that, although under eCO₂, root-level APase did not change,

higher amounts of longer and finer roots proportionally increased APase per surface, which may be a further adaptation to mine Po sources stored in the litter layer under $eCO_2$.

After one year of $eCO_2$, soil-based fine root biomass did not change (Fig. S7), but in contrast to litter-based roots, their productivity significantly decreased by 80% (Figs. 1a, 3a; $Z = 1.90$, $P = 0.05$), accompanied by an increase in root tissue density (RTD; Figs. 1a, 3c; $Z = 2.40$, $p = 0.01$), and by a 117% increase in fine root AMF colonization rate from 25.7% to 56% under $eCO_2$ (Figs. 1a, 3b; $Z = 3.13$, $p = 0.001$). The strong decrease in root productivity, together with the lack of change in the root biomass and the higher RTD, suggests an increase in the soil-based root lifespan under $eCO_2$. This adjustment could provide mechanical protection for mycorrhizal symbionts and against herbivory, suggesting that soil-based roots may have shifted towards a more conservative and collaborative root life strategy[40].

Our results suggest that under $eCO_2$, plant communities adapted the root nutrient acquisition strategies in different ways to target the predominant P pool in the litter and soil to enhance nutrient acquisition efficiency. In the litter-based roots, $eCO_2$ fostered a 'do-it-yourself' nutrient acquisition strategy, with plants investing more in long, thinner, and more branched roots that have a higher potential to access Pi and additionally mineralize Po substrate resources stored in decaying organic matter[19]. In contrast, the shift in soil-based root trait combinations in response to $eCO_2$ suggests that plants are 'outsourcing' nutrient acquisition by fostering mutualistic partnerships with AMF symbionts, to increase the soil volume explored and taking Pi sources through mycorrhizal hyphae instead of producing long and fine roots[17,24].

### Interactions of root trait adaptations and soil P dynamics under $eCO_2$

The soil-based root trait combinations under $eCO_2$ were accompanied by a significant decrease in the soil total Po fraction (i.e., the sum of organic $NaHCO_3$ and NaOH extracted P fractions) by 77.6% (Figs. 1b, 4b; $Z = -2.97$, $p = 0.002$), mainly driven by declines in the more recalcitrant Po fraction (i.e., NaOH; Fig. S14c; $Z = -2.93$, $p = 0.003$). Although Po is the largest P fraction in Ferralsol/Oxisol soils[10], it becomes available only after mineralization through microbe or plant root released phosphatase-catalyzed hydrolysis to Pi[41]. However, as in our study, we did not detect changes in root and microbial APase activity (Figs. S9, S18d), the decrease in Po could be attributable to a change in exudates released into the soil. For instance, enhanced exudation of organic/carboxylic acids could mobilize not only inorganic but also organic compounds from the mineral soil matrix, which could concomitantly liberate Po and Pi[42]. Similarly, photosynthates distributed by roots or AMF hyphae could foster phosphate-solubilizing bacteria in the rhizosphere or hyphosphere[43]. Moreover, the decrease in Po could have been caused by other plants or microbial phosphatases or phytases that were not determined in our study[44].

Furthermore, we found a strong effect of $eCO_2$ on the soil extracellular C:P enzyme stoichiometry (Figs. 1c, S19 b; $Z = -4.89$, $p = <0.001$), which was mainly driven by a reduction of the soil enzymes such as cellobiosidase and β-glucosidase (Figs. 1c, S18a, b) that are responsible for hydrolyzing C compounds. These results could suggest a reduced demand for degrading soil organic C compounds. This could, on the one hand, indicate increased labile C exudate inputs to the rhizosphere, reducing the demand to degrade higher molecular compounds by soil microbial communities[45], or also an adaptation to lower root productivity in soil and lower inputs of plant organic matter. The maintenance of a similar, and relative to C-enzymes, greater investment in P-cycle enzymes could indicate an increase in microbial P demand and, in turn, in an enforced plant-microbial competition for P. However, the absence of a corresponding increase in the Pi fraction (Figs. 1b, 4c) together with the unchanged microbial P biomass and other available soil P fractions also indicates that plants could have enhanced their capacity for P uptake under $eCO_2$. This suggests that plant community collaboration with AMF symbionts seems to be the most cost-effective strategy for soil-based roots under this scenario[26], which could be responsible for supporting the plant's aboveground growth observed at the same site[13]. However, likely the Po loss exceeds the increased P demand by plants for the increased production (based on low P concentrations found in tropical forest trees)[46], a fraction of Po could have also become directly or after enzymatic hydrolysis occluded in the residual P pool and become unavailable, or roots and hyphae may have re-allocated P into different soil layers.

In addition to the observed depletion in soil Po under $eCO_2$, one year later, we also observed a significant decrease in total P concentration in decomposing leaf litter, without any change in litter C decomposition (Fig. S10). This suggests that the adaptations of litter-based fine roots to $eCO_2$ may have longer-term consequences for litter P recycling, with subsequent effects on soil P cycling. Despite the different collection times making it difficult to directly link the P dynamics in the soil and litter layer, our results strongly suggest that the plant community's strategy of investing in multiple root nutrient acquisition strategies impacts the P dynamics in the soil and litter layer, with consequences for the plant root and microbial competition.

### Understanding an efficient understory plant phosphorus acquisition under $eCO_2$

The availability of P is a key constraint on plant productivity across the Amazon basin[11,12] and may limit ecosystem-level responses to $eCO_2$. Our findings indicate that understory tree communities can adjust their P acquisition strategies under $eCO_2$, modulating different nutrient acquisition strategies based on the predominant P forms present in litter and soil layers (Fig. 5). In the short term, such responses appear to sustain the plant nutrient demand and overcome natural P limitations. Importantly, we can exclude the possibility that an $eCO_2$-induced shift in species composition drove these patterns, as no species turnover was observed[13]. Instead, the spatial distribution of root strategies between the litter and soil layers may suggest community-level niche adaptation as a strategy to overcome the low nutrient availability[47]. We highlight the critical role of the litter layer in nutrient recycling within the Amazon forest and, more notably, the unique strategy of growing roots in the litter layer to mobilize nutrients[19,22]. The fact that plants are investing in increasing root foraging in the litter layer, followed by a sustained decrease in decomposing leaf litter P concentrations under $eCO_2$, suggests that P mineralization from Po stored in leaf litter could be an effective strategy to meet nutrient demands under enhanced $CO_2$ conditions. Therefore, we hypothesize that a potential positive $CO_2$ fertilization effect at the ecosystem level, sustaining aboveground growth, even if temporarily, will be dependent on the tree's investment in P acquisition from the litter layer.

Interestingly, and in contrast to some previous FACE studies[48,49], we detected a decrease in soil-based fine root productivity, but higher colonization rates with symbiotic AMF. This potential shift in C utilization by investing in AMF may represent a plant strategy to overcome a possible competition with soil microbes for nutrients. However, our results indicate that after at least one year of $eCO_2$ plant-microbe competition for P, the understory community's response to $eCO_2$ in the studied understory forest was not constrained. Our findings demonstrate that $eCO_2$ can drive pronounced changes in plant systems by influencing microbial stoichiometry and consequently nutrient mobilization. This shift has the potential to alter overall ecosystem functioning, with possible consequences for the Amazon forest's C balance. Nevertheless, ecosystem-level experiments are needed to reassess how our findings play out at the ecosystem scale, accounting for diverse plant life strategies, different plant developmental stages, vertical light availability, and potential acclimation effects.

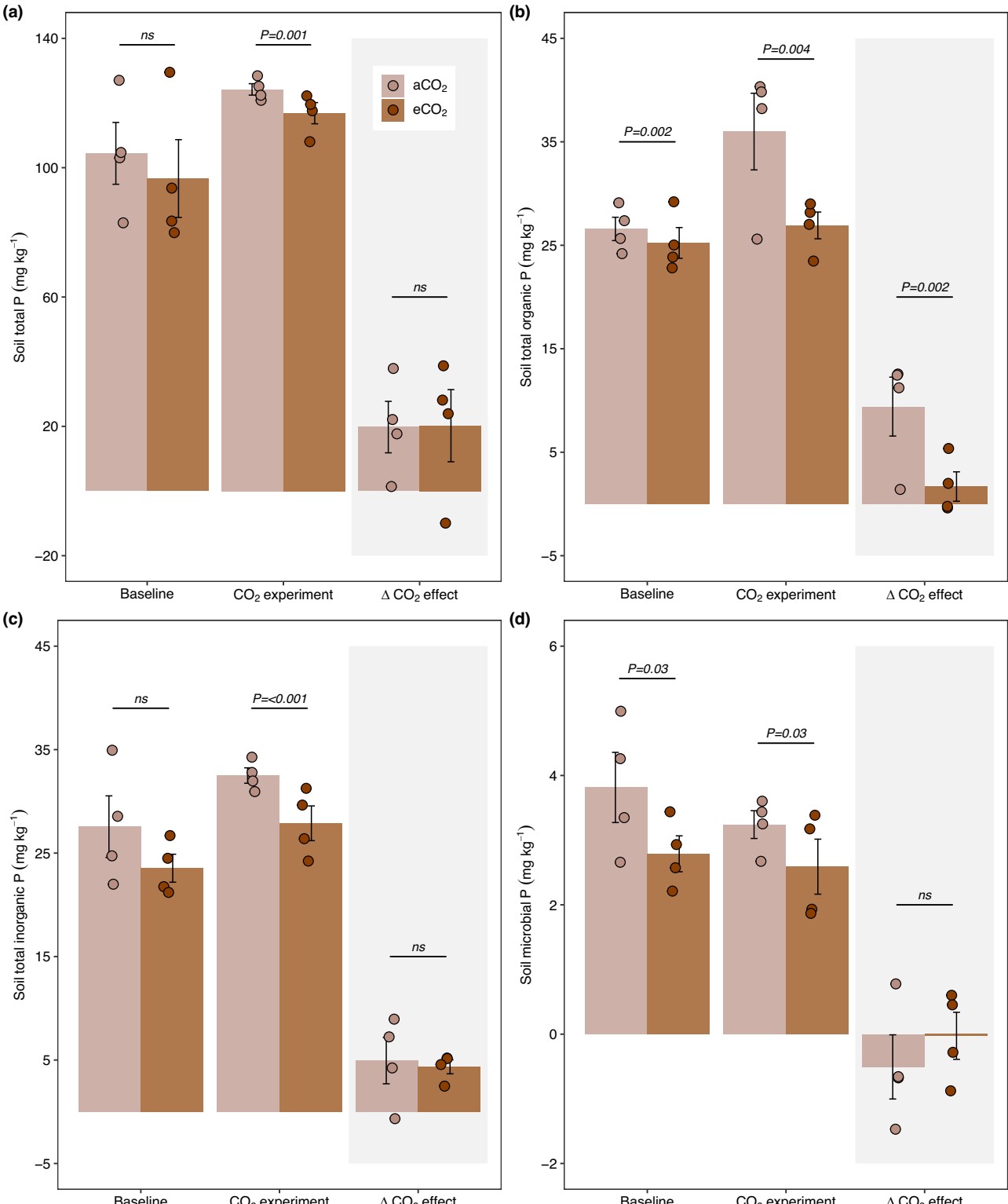

**Fig. 4 | Effects of Elevated CO$_2$ on the soil P budget. (a)** soiltotal P, **(b)** soil total organic P (i.e., the sum of organic NaHCO$_3$- and NaOH extracted P fractions), **(c)** soil total inorganic P (i.e., the sum of inorganic NaHCO$_3$- and NaOH extracted P fractions), **(d)** soil microbial P biomass. We present the baseline collections conducted before the start of elevated CO$_2$, the collection after elevated CO$_2$ was initiated (i.e., CO$_2$ experiment; see more details in Fig. S2), and the delta CO$_2$ effect. The 'delta CO$_2$ effect' was calculated to account for effects caused by spatial and temporal variability as the difference between the "CO$_2$ experiment" and the "baseline" (see "Methods" section). Data were analyzed using generalized linear mixed models, with a specific model developed for each variable. The treatment was considered a fixed factor, and spatial variability was controlled by including the pair of open-top chambers (OTC) as a random factor. The numbers at the top of the bars indicate the probability of no difference between the means for each comparison. The bars represent the mean and standard error of $n = 4$, and the points indicate the samples for each group.

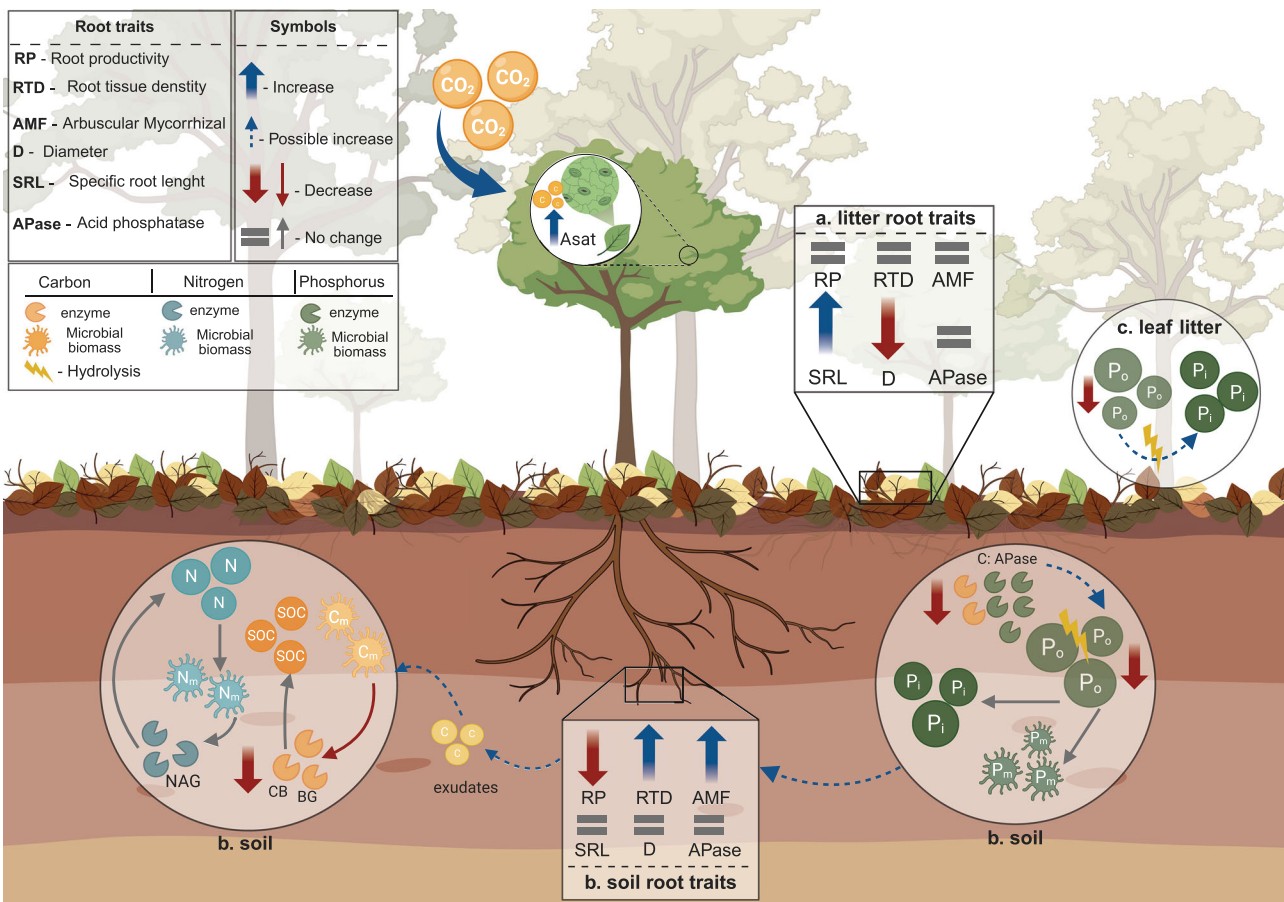

**Fig. 5 | Effect of eCO₂ on plant nutrient acquisition strategies, nutrient concentration in the litter layer and up to 15 cm in the soil, and microbial activity in the soil in a tropical understory community in the Amazon forest.** The belowground process represented was obtained over a two-year timeline under eCO₂ conditions. Where the root traits in the litter layer were obtained after 8 months (**a**), the root traits, nutrient concentration, and microbial activity in the soil after 12 months (**b**), and leaf litter P concentration after 24 months (**c**). Under eCO₂, our results suggest that the plant community modulates different root traits spatially distributed across the litter and soil layers over the collaboration gradient from the root economic space (RES) to increase nutrient acquisition efficiency. Specifically, in the litter layer, the plant community is modulating a "do-it-yourself" strategy, characterized by an increase in SRL and a decrease in root diameter. Furthermore, we observed that the plant community decreased soil root productivity under eCO₂ but increased AMF colonization rate, "outsourcing" nutrient acquisition. Additionally, eCO₂ decreased the soil extracellular enzyme C-to-P ratio, indicating a lower demand for C, which could suggest an increase in P demand driving a plant-microbial competition. This increase in the P demand by both plants and microbes was accompanied by a decrease in the soil organic P (Po) fraction, without any changes in the inorganic P (Pi) or microbial P fraction (Pm). As a later response, our results show a decrease in the leaf litter P concentration without changing the C decomposition, which could suggest a biochemical P mineralization associated with a possible persistent effect of eCO₂ in the litter root traits. The multiple effects of eCO₂ on litter and soil-based root traits are illustrated in the boxes. Blue arrows indicate an increase in response to eCO₂, and dashed blue arrows indicate a possible increase not tested. Red arrows indicate a decrease in response to the eCO₂ effect, and the gray symbol of equal or arrows indicates no change with eCO₂. Created in BioRender. Martins, N. (2026) https://BioRender.com/xo33ejj.

It remains unclear how far the observed shifts in P acquisition under eCO₂ in understory plants can sustain long-term tree growth, as the current P pools are not increasing but are instead being cycled more efficiently between plants and soil. Litter represents a dynamic pool, constrained by plant inputs and outputs, meaning its contribution has a natural limit. Within this constraint, the partitioning of P between soil, plant uptake, and microbial demands becomes critical in determining long-term ecosystem responses[29]. The intensification of soil P cycling observed in this study provides direct evidence that eCO₂ alters P dynamics, and it was associated with an aboveground C sink in the understory. Our findings point to the potential for increasing P supply due to an increase in the P cycling in the Amazon forest, yet do not rule out progressive P limitation of plant productivity with eCO₂, as increasing P demand ultimately exceeds supply. Therefore, given Amazon's central role in sustaining local and global climate and C balance, understanding whether these shifts can sustain long-term forest growth under rising CO₂ is critical.

## Methods

### Site description

The Open-top chambers (OTC) experiment was implemented in the experimental area of the AmazonFACE program (−2° 35' 40.29, −60° 12' 28.69)[33], located in the Central Amazon at the "*Cuieiras*" experimental reserve about 70 km north of Manaus (Amazonas, Brazil; Fig. S1). The study area was located on a plateau with old-growth, lowland Terra-Firme Forest vegetation[50]. The soil is clay-rich (67.7% clay, 19.9% sand, and 12.3% silt) and highly weathered Geric Ferralsols, with a pH of 3.94 and a low concentration of total P and rock-derived nutrients[10]. Average annual rainfall is about 2500 mm, with the wettest period from December to May and the dry season spanning August to September (<100 mm of precipitation). The average temperature ranges between 24 °C and 27 °C[51].

### Experimental design and sampling timeline

Eight polypropylene open-top chambers (OTCs), each with a diameter of 2.4 m and a height of 3 m, were randomly installed in the forest

understory (Fig. S2). To prevent root proliferation and other effects from plants outside the OTCs, each chamber was surrounded by a circular soil trench measuring 30 cm in width and 50 cm in depth. The chambers were arranged in pairs, with four control chambers maintained at ambient $CO_2$ (a$CO_2$) and four elevated $CO_2$ (e$CO_2$) treatments, in which $CO_2$ concentration was increased on average by ~ 300 ppm relative to the respective ambient OTC. Inside each chamber, concentrations of $CO_2$ and water vapor ($H_2O$) were measured using a $CO_2$/$H_2O$ non-dispersive infrared (NDIR) gas analyzer (LI-840A, Li-Cor Biosciences, Lincoln, NE, USA) connected to a central logger system (Campbell Scientific CR1000 data loggers). The $CO_2$ was injected into the e$CO_2$ chambers through a gas line connected to a central cylinder system and distributed by fans close to the injection hose. The injection of $CO_2$ was carried out during daylight hours (from 6 a.m. to 6 p.m.) in the e$CO_2$ OTCs when the difference between the treatment pairs was less than 200 ppm. The $CO_2$ injection in the e$CO_2$ chambers starts on 1 November 2019. The e$CO_2$ system was adjusted and stabilized during the first two months (i.e., November and December 2019), and after this period, it maintained an average increase in $CO_2$ concentration of ~300 ppm until October 2021 (Fig. S4; a$CO_2$: $455.14 \pm 0.62$; e$CO_2$: $764.34 \pm 2.96$).

Sample collection occurred in two phases (Fig. S3). The first baseline phase (2019) encompassed all preparatory procedures, including OTC installation and an initial sampling campaign. We installed ingrowth cores in the litter layer and in the soil in February 2019 to study baseline root productivity and collected them in August 2019. Also in August 2019, we conducted a first soil sampling campaign to characterize soil nutrient concentrations, microbial biomass, and activity rates. Ingrowth cores were reinstalled in November 2019, immediately before starting the $CO_2$ enrichment. The e$CO_2$ phase started in November 2019. Ingrowth cores to investigate litter-based fine-root productivity were assessed after eight months (November 2019 to July 2020); soil-based fine root productivity was assessed by collecting ingrowth cores after twelve months (November 2019 to November 2020). Additionally, we collected soil samples in November 2020 after twelve months of e$CO_2$ to investigate root biomass, soil nutrient concentrations, microbial biomass, and enzyme activity. A litter-decomposition experiment was initiated in September 2020 and was concluded with a single harvest in September 2021. When available, baseline data were used to minimize a possible natural spatial variability of variables between OTC and seasonal effects.

## Soil and plant characteristics

The OTC design enables the monitoring of plants from the understory. Therefore, we are considering plants with a maximum height of 3 meters. Furthermore, we observed variations in height averages, ranging from 0.28 to 3 m, and in diameter at the base (DB), ranging from 3.9 to 35 mm[13]. The number of plants inside the OTCs varied from 10 to 18; a total of 56 species, belonging to 26 distinct families, were identified among the eight OTCs, comprising plant individuals. More information about the plant species distribution across OTCs and aboveground responses can be found in Damasceno et al. (2024). Soil chemical and physical properties inside the OTCs follow the same pattern previously described for the Ferralsols in Central Amazonian[10], up 15 cm deep, the soils present an average of 102.58 mg kg$^{-1}$ of total P, where 47% is P residual and 52% is extractable P, 3.7 of pH, 3.06% of carbon, and 0.24% of nitrogen (see details by OTC in Table S1).

## Root productivity

**Litter layer.** Litter-based fine root productivity was monitored by installing two rectangular-shaped ingrowth litter traps ($20 \times 15$ cm) per OTC in February 2019 (placed horizontally at the mineral soil surface and filled with local litter). The first collection happened after 6 months (baseline collection; August 2019; Fig. S2) and 1 week before starting the $CO_2$ injection (i.e., November 2019); all the ingrowth litter traps were zeroed by removing any roots found inside them. The initial plan was to collect all the ingrowth cores (i.e., litter and soil) in an interval of three months[52]. However, the outbreak of the global pandemic of SARS COVID-19 did not allow us to maintain the regular fieldwork schedule, and the collections were adapted as much as possible. The harvest of litter-based ingrowth cores happened 8 months after starting the e$CO_2$ treatment (July 2020; Fig. S2). Litter-based fine root productivity (expressed in mg$^1$ cm$^{-2}$) was determined by carefully collecting all roots from the two ingrowth litter traps by OTC that were in direct contact with the litter layer. Annual litter-layer fine root net productivity (mg cm-2 year$^{-1}$) was calculated by dividing roots produced by the number of days of the collection interval and multiplying daily rates by the total number of days in the year. Subsamples of roots were used to determine morphological parameters and phosphatase activity.

**Soil.** Soil-based fine root productivity was measured by installing four 12 cm diameter, 15 cm-deep, root-free ingrowth cores per OTC. For collecting baseline data, we installed ingrowth cores in February 2019, and similar to litter-based roots, we collected ingrowth cores in August 2019 (Fig. S2). One week before starting the e$CO_2$ treatment, all ingrowth cores were emptied and zeroed (i.e., all roots were removed). Here, we present the data collected after twelve months of e$CO_2$ (i.e., November 2020). However, considering that the ingrowth cores were emptied in August 2020, we have the root that was produced over the three-month period. To account for spatial variability within the OTC, we used two ingrowth cores per OTC to quantify root productivity and morphological parameters of fine roots. Two additional cores were used to measure fine root phosphatase activity and mycorrhizal colonization. We considered fine roots smaller than 1 mm in diameter to focus on absorptive roots (first, second, and third orders)[53]. To determine soil fine root productivity, we manually collected all roots in the field immediately after each sampling, for 12.5 min, and then in five intervals of 2.5 min (as an adaptation of the time interval methodology)[52]. The collected roots were washed and cleaned gently using a brush to remove soil particles. The root-free soil was returned to the ingrowth core sampling spot. We selected the logarithmic curve that provided the best model fit to represent the total root weight and then calculated the soil fine root productivity. Soil-based fine root productivity was calculated using a three-month collection interval from August to November 2020 and is expressed as the dry mass of roots per day and total area of the ingrowth core (mg cm$^2$ day$^{-1}$). We calculated the soil-based root productivity by year (i.e., mg cm-2 year$^{-1}$) as described before for the roots in the litter layer.

## Fine root morphology parameters

All litter-based fine roots and subsamples of soil-based fine roots collected from the ingrowth cores (collected during the first 12.5 min time interval, representing more than 50% of the total sample) were scanned in high resolution (600 dpi) and analyzed using WinRHIZO (WinRHIZO Regular, Regent Instruments, Canada) to determine fine root morphological traits. We determined specific root length (SRL), specific root area (SRA), root tissue density (RTD), and mean root diameter. Then, samples were dried for 72 h at 65 °C to determine dry root mass[54]. SRL (m g$^{-1}$) was calculated as fine root length per unit root dry mass, SRA (cm$^2$ g$^{-1}$) as fine root surface area per unit dry mass, and RTD (g cm$^3$) as root dry mass per unit root volume.

## Root phosphatase activity

Subsamples of litter-layer and soil-based fine roots were analyzed for root surface potential acid phospho-monoesterase activity (APase) using an adapted fluorometric microplate assay protocol[54–56]. From each ingrowth core, approximately 10 mg cleaned root subsamples were incubated gently shaking with sodium acetate buffer and methylumbelliferyl-phosphate (MUF) substrate in 2 ml snap-cap vials

for 30 min. A further root subsample was incubated with sodium acetate buffer only as a negative control. The reaction was stopped by adding 50 µl of 1 M NaOH to all samples, negative controls, and methylumbelliferyl standards; then, slurry aliquots were pipetted in triplicate into a black 96-well microplate. After 20 min, the fluorescence was read at 365 nm excitation and 450 nm emission (ref. Tecan Infinite®). Subsequently, these fine root subsamples were scanned and dried at 65 °C for 72 h. Potential root phosphatase activity was expressed in $\mu mol\ mg^{-1}$ root dry mass $h^{-1}$.

## Arbuscular mycorrhiza colonization

For litter-based roots, we determined AMF colonization rates of roots using subsamples of fine roots inside the litter bags placed within the OTCs to monitor leaf litter decomposition (see more details on the litter decomposition design below). For soil-based roots, we used subsamples collected from soil ingrowth cores up to 15 cm in depth, from which roots were collected to calculate productivity. After cleaning, the segments of fresh absorptive roots from the first three orders were stored in 50% ethanol. The process of clearing and staining was previously described for tropical roots[54,57]. Initially, the roots were cleared in a 2.5% potassium hydroxide (KOH) solution and autoclaved at 120 °C for approximately 10 min. Subsequently, the roots were placed in an alkaline hydrogen peroxide ($H_2O_2$) bleaching solution for 30 min and then acidified using a 2% hydrochloric acid (HCl) solution for an additional 30 min. Then, the roots were immersed in 0.05% Trypan Blue solution until a consistent blue coloration was achieved. Uniformly stained root fragments were randomly selected from the subsample to quantify the extent of colonization by AMF by the cross-section method[58]. These fragments were mounted on slides and subjected to a high-resolution optical examination at 10x magnification, enabling the determination of the total length of roots colonized by AMF fungi.

## Leaf litter decomposition

We conducted a one-year leaf litter decomposition experiment inside the OTCs starting in September 2020. We chose to use leaves of *Vismia sp.* collected from a single tree that had recently fallen after a heavy storm in the area close to the experimental site to reduce the chemical variability of litter material among OTCs. The collected leaves were dried at 65 °C for 72 h and stored dry. We prepared five litter bags with a 2 mm mesh size (20 × 20 cm) per OTC, each containing 10 g of dry leaves. After 1 year, all litter bags were collected, remaining leaves were weighed fresh, and then dried at 65 °C for 72 h to determine a dry weight-to-fresh weight ratio, which was used to calculate the total litter dry weight. Leaf litter mass loss (ML) was calculated as a percentage of the total initial mass as in Eq. (1):

$$ML = ((m_0 - m_1)/m_0) \times 100) \qquad (1)$$

With $m_0$ representing the initial litter dry weight and $m_1$ representing the litter dry weight at the collection.

The remaining litter was milled to fine powder and analyzed for total carbon (C), nitrogen (N), P. Total C and N were analyzed in an automatic CN analyzer (Vario Max CN, Elemental Analyzer, Germany) by mass spectrometry. Nitro-perchloric acid digestion was used to determine the total concentration of P[59]. Total P was read on a UV Spectro-photometer (Model 1240, Shimadzu, Kyoto, Japan). Nutrient concentration is shown as g of nutrient per kg of dry leaves (g $kg^{-1}$).

## Soil fine root biomass, microbial activity, and nutrient concentration

The soil nutrient concentration and microbial activity baseline were conducted in two different periods in 2019 (February and August; see Fig. S2 for more details). After 1 year with eCO₂ (i.e., November 2020) in the OTCs, we collected two soil samples per OTC to quantify fine root biomass stocks, microbial activity, nutrient concentration, and P

fractions in response to eCO₂. Soils were sampled 15 cm deep using an auger (ø 5 cm) and then transported to the thematic laboratory of soils and plants (LTSP) at the National Institute of Amazonian Research (INPA) to sort the roots to determine the root biomass and sieve the fresh soil through a 2 mm mesh size for further microbial activity analyses.

**Soil root biomass stock.** Fine root biomass was determined from the collected soil cores following the abovementioned time interval methodology[52]. After sorting, all roots were washed and divided into different diameter classes (<1 mm, 1–2 mm, and >2 mm) and dried at 65 °C for 72 h. We used only the roots considered acquisitive (<1 mm in diameter) to better compare with the fine root productivity rates. Root biomass was calculated using the area of the auger and the soil depth and expressed in $mg^1\ cm^{-2}$.

**Soil microbial biomass.** We determined microbial biomass C, N, and P using the chloroform fumigation extraction method of fresh soil within 72 h after soil stock collection[60]. From each sample, 2 g of soil was fumigated with chloroform for 24 h. Then, the samples were divided into two subsamples (1 g each) for extractions with 20 ml 1 M KCl to analyze organic extractable C and total extractable N, and another for extractions with 20 ml of 0.5 M NaHCO₃ (pH 8.5) to analyze total extractable P. At the same time, another set of samples was used for extraction with the same extractors without chloroform fumigation. Fumigated and non-fumigated extracts of KCl and NaHCO₃ were analyzed[19], and microbial C, N, and P were estimated as the difference between concentrations in the fumigated and non-fumigated extracts and expressed by soil dry mass.

**Soil microbial extracellular enzyme activity.** We measured the potential activity of four different soil extracellular enzymes involved in C, N, and P cycling (cellobiosidase: 4-MUF-cellobioside (CB), β-glucosidase: 4-MUF-β-D- glucopyranoside (BG), N-acetylglucosominidase: 4-MUF-β-D-glucosaminide; (NAG) and acid phosphatase: methylumbelliferyl-phosphate; (APase). Briefly, we homogenized 0.5 g of soil and 50 ml of 100 mM sodium acetate buffer (pH 5.5) by vortexing samples for 1 min. Per sample aliquots of 200 µl slurry were pipetted into 96 black microplates in triplicate, and incubated with the respective substrates for 40 min. In addition, we included substrate and quenching blanks in triplicate, as well as methylumbelliferyl standard curves on all microplates[55,61,62]. The fluorescence was determined on a microplate reader (TECAN i-control 200Pro, Groedig, Austria) at an excitation wavelength of 365 nm and an emission wavelength of 450 nm. Potential extracellular enzyme rates are expressed in nmol $g^{-1}$ dry soil $h^{-1}$. We calculated microbial CNP ratios as an indicator for potential microbial nutrient limitation or preferential mining strategies. We used the log of C:P enzyme ratio (log (CB + BG) / log APase), the N: P enzyme ratio (log NAG/ logAPase), and normalized C (sum of CB and BG), N (NAG) and P acquisition (APase) to microbial biomass C[63,64].

**Soil element concentrations.** Total soil C and N concentrations were analyzed on an elemental analyzer (EA 1100, CE Instruments, Milan, Italy) coupled to a Finnigan MAT Delta Plus IRMS (Thermo Fisher Scientific, MA, USA). Total soil P concentrations were analyzed using concentrated sulfuric acid ($H_2SO_4$, 18 $M$) digestion followed by the addition of 30% $H_2O_2$[65]. Furthermore, we analyzed the soil for different inorganic and organic P fractions using sequential extractions[10,66]. First, resin extractable P in water is extracted, followed by 0.5 M NaHCO₃ (pH 8.5, bicarbonate fraction), 0.1 M NaOH (hydroxide fraction), and 1 M HCl (hydrogen chloride fraction). All extracts were analyzed for inorganic Pi; additionally, the NaHCO₃ and NaOH extracts were further digested with Na-persulfate to get the respective organic P fractions. This yielded a total of six different fractions: resin-P, bicarbonate-Pi, bicarbonate-Po, NaOH-Pi, NaOH-Po, and HCl-P. All six P-fractions, as well as the total soil P

were analyzed for $PO_4$ concentrations photometrically, the results are given in mg kg$^{-1}$ dry soil. The sum of the six P fractions was used to calculate soil extractable P. The residual P fraction is the difference between total soil and extractable P[20].

## Data analysis

All statistical analyses were performed in R version 4.5.2[67]. To minimize the possible natural spatial variability between chambers and seasonal effects, we calculated a delta $CO_2$ effect (delta $CO_2$) as the difference between the measurements after starting the $CO_2$ fumigation and baseline measurements (i.e., a$CO_2$ treatment−a$CO_2$ baseline, and e$CO_2$ treatment−e$CO_2$ baseline). The "difference-in-difference" approach[68] isolates the within variation for both the a$CO_2$ and e$CO_2$ chambers, putting the gains or losses on the same scale. For all variables for which a baseline collection was available (see Fig. S3 for details), the delta $CO_2$ was calculated considering the difference between the respective OTC and samples at both times. For those variables for which obtaining a baseline collection was impossible, the effect of elevated $CO_2$ is represented by the respective collection after the start of the $CO_2$ fumigation. A statistical analysis was conducted for each separate collection (baseline, after-start e$CO_2$, and delta $CO_2$ effect) to understand the dynamics of the variables before and after the increase in $CO_2$ concentrations (see Supplementary information).

In the main text, we consider the delta $CO_2$ as the real effect of the increase in $CO_2$ concentrations on soil and litter fine root dynamics, soil microbial activity, and soil nutrient concentration. General linear mixed models (GLMM) using the glmmTMB package[69] were used for each variable to test the difference between the treatments (a$CO_2$ and e$CO_2$) as fixed factors and the pairs of OTCs (see details in Table S1) as random factors to control for natural environmental and spatial variability. In these cases, the samples collected at each OTC were averaged, and the models were run using the OTC ($n = 4$ by treatment) as replication for all variables. We fitted the model using the Gaussian statistical family as the basis for all variables, and when model diagnostics indicated inadequate fit, we tested other families (e.g., Tweedy and gamma). The assumptions for normality and homogeneity of the residuals using the simulateResiduals function from the DHARMa package[70]. In the main text, the results are reported graphically as the effect size of delta $CO_2$ (i.e., delta e$CO_2$ per delta a$CO_2$) calculated with Cohen's d function by the effect size package[71].

## Reporting summary

Further information on research design is available in the Nature Portfolio Reporting Summary linked to this article.

## Data availability

The datasets generated in this study have been deposited in the Zenodo repository under accession code https://doi.org/10.5281/zenodo.18495501.

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

## Acknowledgments

The authors thank the AmazonFACE Program, the Brazilian Ministério de Ciência, Tecnologia e Inovação through its Fundo Nacional de

Desenvolvimento Científico e Tecnológico - FNDCT, the UK Government's Foreign, Commonwealth and Development Office, the LBA Program, the National Institute for Amazonian Research (INPA), the Biogeochemical Cycles Laboratory (INPA), and all their collaborators for the general support in carrying out this study. This research was funded by the Serrapilheira Institute (Grant 1708-15574), with additional funds from Brazil's Coordination for the Improvement of Higher Education Personnel (CAPES) (Grant CAPES-INPA/88881.154644/2017-01 and grant 23038.007722/2014-77). N.P.M. acknowledges funding support from Coordination for the Improvement of Higher Education Personnel (CAPES)—Finance Code 001, Brazilian Biodiversity Fund (FUNBIO) grant 2019, and Humboldt Research Fellowship from the Alexander von Humboldt Foundation. L.F. was supported by the European Research Council Synergy grant #610028 Imbalance-P and by the European Union's Horizon 2020 research and innovation program under the Marie Sklodovska-Curie grant agreement No 847693 (REWIRE). L.F.L and A.R would like to thank the financial support of the Bayerische Staatskanzlei through the grant associated with the Amazon-FLUX project. F.B.B is continuously supported by the Brazilian National Council for Scientific and Technological Development (CNPq) grant #312878/2023-0. T.F.D acknowledges the financial support from the Brazilian National Council for Scientific and Technological Development (CNPq) grants 312589/2022-0 and FAPESP grants 2024/02277-4, 2024/08543-8 and 2024/16357-0. D.M.L was funded by FAPESP (grant 2023/09046-5) and CNPq (grant 305807/2024-2). FH gratefully acknowledges funding from IIASA and the National Member Organizations that support the institute. CAQ acknowledges the financial support from the Brazilian National Council for Scientific and Technological Development (CNPq) grant 312866/2021-6.

## Author contributions

N.P.M., L.F. and C.A.Q. planned the study. N.P.M., L.F.L., I.A., B.N.S.B., C.C.S.S., C.C., R.D.P., A.D., V.R.F., S.G., A.G., J.G.M., A.C.M.M., A.C.M., L.R.O., I.S.P., M.P.M, Y.R.S., B.T. and G.U. collected the data and/ or helped with project logistics. N.P.M., L.F., C.C., C.P., G.R., J.S.R., Y.R.S., L.S.S. and G.U. conducted and/ or helped with the laboratory analyses. N.P.M. C.A.Q. and D.M.L. wrote the grants that funded this research. N.P.M., L.F. and F.B.B. conducted the statistical analysis. N.P.M. and L.F. wrote the manuscript. C.A.Q., L.F.L., O.J.V.B., R.J.N., I.P.H., F.B.B., T.F.D., K.F., F.H., D.M.L., A.R. and F.D.S. contributed with the manuscript reviews.

## Funding

## Competing interests

The authors declare no competing interests.

## Additional information

Nathielly P. Martins ®[1,2] ✉, Lucia Fuchslueger ®[3] ✉, Laynara F. Lugli[2], Oscar J. Valverde-Barrantes[4], Richard J. Norby ®[5,6], Iain P. Hartley ®[7], Izabela Aleixo ®[8], Fabricio B. Baccaro ®[9], Barbara N. S. Brum ®[2], Crisvaldo Cássio Silva de Souza[2], Carine M. Cola ®[2], Raffaello Di Ponzio ®[10], Amanda Damasceno[2], Tomas F. Domingues ®[11], Vanessa R. Ferrer[2], Katrin Fleischer[12], Sabrina Garcia ®[2], Alacimar Guedes[2], Florian Hofhansl ®[13], David M. Lapola ®[14], Juliane G. Menezes[2], Anna C. M. Moraes ®[2], Ana Caroline Miron[15], Leonardo Ramos de Oliveira[2], Cilene Palheta[2], Iokanam S. Pereira[2], Maria Pires Martins[2], Gyovanni Ribeiro[2], Jéssica Schmeisk-Rosa[2], Anja Rammig ®[1], Flavia D. Santana ®[2,16], Yago R. Santos ®[2], Lara Siebert Silva[2], Bruno Takeshi T. Portela[2], Gabriela Ushida[17] & Carlos A. Quesada[2]

[1]Professorship of Land Surface–Atmosphere Interactions, TUM School of Life Sciences, Technical University of Munich, Freising, Germany. [2]Coordination of Environmental Dynamics, National Institute for Amazonian Research, Manaus, Brazil. [3]Center for Microbiology and Environmental Systems Science, University of Vienna, Vienna, Austria. [4]Department of Biological Sciences, Institute of Environment, International Center of Tropical Botany, Florida International University, Miami, Florida, USA. [5]School of Geography, Earth & Environmental Sciences, University of Birmingham, Edgbaston, UK. [6]Environmental Sciences Division, Oak Ridge National Laboratory, Oak Ridge, TN, USA. [7]Geography, Faculty of Environment, Science and Economy, University of Exeter, Exeter, UK. [8]Coordination of Technology and Innovation, National Institute for Amazonian Research, Manaus, Brazil. [9]Department of Biology, Federal University of Amazonas, Manaus, Brazil. [10]Programa de Pos-graduação em Ecologia, Conservação e Manejo da Vida Silvestre, Instituto de Ciências Biologicas, Universidade Federal de Minas Gerais, Belo Horizonte, Brazil. [11]Departamento de Biologia, Faculdade de Filosofia, Ciências e Letras de Ribeirão Preto, University of São Paulo, Ribeirão Preto, SP, Brazil. [12]Systems Ecology, Amsterdam Institute for Life and Environment, Vrije Universiteit Amsterdam, Amsterdam, The Netherlands. [13]International Institute for Applied Systems Analysis, Laxenburg, Austria. [14]Universidade Estadual de Campinas – UNICAMP, Campinas, Brazil. [15]Department of Biology, University of Hamburg, Hamburg, Germany. [16]Chico Mendes Institute for Biodiversity Conservation (ICMBio), Brasília, Brazil. [17]Forest Ecology and Forest Management, Wageningen University & Research, Wageningen, The Netherlands. ✉e-mail: nathielly.martins@tum.de; lucia.fuchslueger@univie.ac.at

