## [Peer Review File · Nature Communications]

Amazonian understory forests change phosphorus acquisition strategies under elevated CO₂

Corresponding Author: Dr Nathielly Martins

Version 0:

Reviewer comments:

Reviewer #1

(Remarks to the Author)

The Ms by Martins et al., tried to explore the impact of eCO₂ on P acquisition strategies in Amazonian understory tree communities, addressing the role of root morphological changes and AMF in overcoming P limitation. Upon reading the relevant literature and current MS by Martins et al., I could say this study is timely and of high ecological relevance, particularly in the context of nutrient-limited tropical forests facing climate change. The experimental design using OTCs in P-depleted Amazonian soils is robust and innovative. However, the MS currently suffers from critical conceptual, structural, and mechanistic weaknesses, and the interpretation at times outpaces the data presented. Significant rewriting, clarification of causality, and broader ecosystem framing are required. Here are few of my comments/suggestions.

Major comments

- The experiment was carried out with minimum treatments and with 4 replicates. Given the small spatial and temporal scale of the OTC setup, how do the authors justify statements about Amazon-wide carbon sink resilience and ecosystem functioning? As reported in the abstract. “decisive to the Amazon forest’s resilience”.
- There needs some more details about soil samples i.e., depth of soil sampling or soil profile characterization? This is important for nutrient and texture context, how often measurements of root traits, soil nutrients, or other variables were taken relative to the CO₂ treatment timeline?

Abstract

- Decisive to the Amazon forest’s resilience..... these claims feel like stronger and more overestimated. Authors may choose soften speculative because there is lot to determine and analysis before going to final decisions.
- Capacity to act as a future carbon sink..... this is well-known fact that the forest already acts as a carbon sink, authors should change words like future role or continued capacity.
- I don’t know why the authors used citations in the abstract. References should be omitted from the abstract and there are some nonscientific/ inappropriate words, they should be replaced by appropriate words. Here are a few examples, spatial segregation, less productive.
- Please provide abbreviation with arbuscular mycorrhizal fungi, as AMF etc and results should be written in the past tense.
- Main text.. should be replaced by Introduction, please follow the journal guidelines.
- Authors sometimes used phosphorous, and sometimes abbreviation P, please be consistent when using abbreviation and full name, please check throughout MS.
- 57-59 It sounds like.....nutrient feedbacks and experimental data are not accounted for in Earth system models,” which is confusing. Please add subject of ...not accounted for..... How do current Earth system models implement or fail to implement nutrient feedback loops, especially P and N limitations?
- L59 suggest,.....please remove comma
- 69-71, very long sentence and too much general information. Please split it and concise information
- Should be ...microbial communities, rather than microbial community
- different predominant P forms available?? What are the predominant P forms, please be specific.
- Some words are not defined earlier, for examples, trait syndromes, root economic space,, collaborative axis...
- What is the relative impact of acid phosphatase activity vs. mycorrhizal uptake in Pi acquisition under elevated CO₂? What proportion of P uptake in Amazon understory comes from litter vs. mineral soil under eCO₂? Please provide explain for better understanding the background.
- L91 Does “it” mean to Pi or organic P?, please be specific
- plants increased photosynthetic capacity but without changing 100 aboveground biomass growth.... Please avoid repetition, as explained previously, I suggest to add possible

mechanisms for belowground response or how hyper-diversity could change dynamics..

- Eucalyptus should be italicized and provide full scientific name if authors are referencing plant, *Eucalyptus globulus*
 - at the study site.... Please add the site name.
 - Strikingly.. please avoid non-scientific words
 - 114, were set up in pair ... here authors meant four pairs???
 - we aim.. should be we aimed to...
 - whether the authors randomize the chamber placement or account for spatial gradients light, slope etc. please also mention this.
 - How representative are these understory tree species of the broader Amazon understory in terms of function and diversity?
 - ,,,,under eCO₂, On the one hand.. . please remove comma after eCO₂
 - Authors used two different terminologies, soil-based roots, roots in the soil, please be consistent with the use of terminologies.
 - a trade-off between root production and symbiosis..... what does this trade-off suggest.....?
 - 126 arbuscular mycorrhizal (AMF) colonization.... should be arbuscular mycorrhizal fungi (AMF).....please use abbreviations once full name is used. Please check throughout MS.
 - roots are favoring..... roots don't favor strategies,..... plants do. Please change
 - once you used full name, root phosphatase activity, please abbreviate next.....
 - 148 In summary, in litter-based roots....please avoid concluding after every paragraph.
 - Please use abbreviations for phosphorus.
 - A 189% reduction is not mathematically logical unless it states more than complete loss. Perhaps meant reduction to 11% (89% decrease) or activity decreased by 89%. Please double check the data.....
 - You make several causal claims (e.g., eCO₂ leads to enhanced P mobilization) that are well-supported I suggest to explicitly framed as interpretations, not facts.
 - 174-178-- last sentence is quite long and multiple arguments at once,,, please avoid conclusion after every paragraph.
 - 170-172 enzyme allocation toward Pit should be functional P limitation
 - Is it possible that microbial community composition changed (even if microbial P pool didn't)?
 - Please use abbreviations AMF, P, eCO₂ etc please be consistent with the use of elevated eCO₂ or enhanced eCO₂,
 - progressive P limitation.... What is the cause of it and please also add explanation.
 - future research should evaluate whether these root and microbial adjustments.... authors provided too many general future directions that is somehow is not possible, and no practical use, I suggest to provide simple ones, and practical, and feasible.
 - References have lot of issues,
 - Authors sometimes used et al., and sometimes used all the authors name, please be consistent
 - Don't need to capitalized first letter of the title of the paper in the reference
 - In, CO₂2 should be subscript.
 - Authors used lot of references, even cited for general statement, please reduced number of references.
 - Tweedie distribution family, it should be Tweedy seem like typo
 - Some sentences are quite long and complex. Breaking them into shorter sentences
 - Use past tense consistently for methods already performed.
 - When introducing abbreviations i.e., AMF for arbuscular mycorrhizal fungi), define once and use thereafter.
- I suggest to add a map of the study area, so that it would be clear how many points, locations and area was covered on the actual map.
- In Table, S2 , don't need to italic some of the words. i.e, Baseline Elevated CO₂ experiment Delta CO₂ effect in Table S4, please subscript of the poweres of the chemical names, also check throughout Ms... NaHC3... NaHCO₃ there is contradiction between main text and Figure S1. 3 m height and 2.4 cm diameter, 2.4 cm is quite small diameter, please double check.
- Figure S3, The data represent the period between January 2020 and October 2021, samples was done just after peak of Covid-19, there were some challenges in taking readings, climate factors, etc
- Figure S3, some texts are blurred, difficult to read. please increase Quality

Reviewer #2

(Remarks to the Author)

This manuscript provides timely experimental evidence on how Amazonian understory plants adapt root P-acquisition strategies under elevated CO₂. The in situ OTC experiment in P-depleted soils addresses a critical gap in understanding tropical forest resilience to climate change. The dual-strategy mechanism (litter "DIY" vs. soil "outsourcing") is compelling and advances ecological theory. While the findings are well-supported, however, I have a few comments regarding some methodological limitations and mechanistic ambiguities to enhance logical flow and clarity.

1. Statistical Power and Replication

With only n=4 OTCs per treatment, statistical power is limited despite GLMMs accounting for spatial variability.

2. Mechanism of Soil Organic P Decline

The 78% reduction in soil organic P (Fig. 1c) is mechanistically ambiguous:

Not linked to root/microbial APase activity (Fig. S8/S19b).

Attribution to "reduced organic P input from litter" (L162) contradicts observations: Litter-layer roots mobilize P, which should increase soil P inputs.

Recommendation:

Test correlations between litter-root traits (SRL, diameter) and soil P decline.
Discuss alternative drivers (e.g., microbial priming, unmeasured phosphatases).

3. Microbial Competition Narrative

Reduced C-degrading enzymes (CB, BG) suggest microbial P limitation (L167–172), yet:
Microbial biomass P (P_m) did not change (Fig. S17c).
AMF colonization increased (outsourcing strategy), implying reduced plant-microbe competition.

4. Transient vs. Sustained P Mobilization

Litter-layer P mining may be short-term effective but unsustainable:
Litter P is finite and input-constrained (L211–212).

Some minor comments:

Lines 428–429: Root Productivity Metrics: Unify terminology (e.g., " $\text{mg cm}^{-2} \text{day}^{-1}$ " vs. " mg year^{-1} "; Methods P14–15).

Line 143: Phosphatase Activity: Resolve discrepancy: Fig. 1a lists "APase" with p-values, but text states no change.

Line 154: Litter Decomposition: Discuss why mass loss was unaffected despite P decline.

Line 126: AMF Colonization: Report absolute % colonization (not just % change;).

Figures & Tables

Fig. 1: Specify error bars (95% CI or SE? L335); include $n=4$ in caption.

Extended Data: Add schematics of OTC/trench design (Fig. S1) to Methods.

Additional Suggestions:

Supplemental Analyses: Correlate root trait shifts with soil P fractions.

Give a figure about P cycling or P budget in different layers among soil, litter and trees.

Table S1 should describe proportion of AMF and ECM trees.

Future focus: What is the mechanism of mycorrhizal plants allocating C within their roots and mycorrhizae? How to accurately distinguish the contribution of the direct pathway via roots and the mycorrhizal pathway via AM fungal hyphae to plant P acquisition?

Version 1:

Reviewer comments:

Reviewer #1

(Remarks to the Author)

This MS provides robust experimental evidence that elevated CO_2 can reshape phosphorus acquisition strategies in Amazonian understory forests through coordinated changes in root traits, AMF, and soil P cycling. The work is original, methodologically sound, and of broad relevance to understanding tropical forest resilience under climate change. I reviewed the revised Ms and also read other reviewer comments, I found that authors revised the Ms and address all of the comments. Now quality of Ms is significantly improved. I therefore recommend acceptance of this Ms.

Comment; In the abstract, please use the subscript function in the MS word software for CO_2 , check throughout Ms.

Reviewer #2

(Remarks to the Author)

I appreciate the authors' detailed rebuttal and the substantial additional analyses. All of my major concerns—(i) statistical power justification, (ii) mechanistic explanation for the 78 % soil- P_o decline, (iii) clarification of the "microbial competition" narrative, and (iv) acknowledgement of temporal constraints on litter-P mining—have been satisfactorily addressed. The manuscript now presents a balanced discussion and provides the data required to support its conclusions. I support publication after minor revision.

Essential revisions (no new experiments required)

1. Statistical transparency

Table S1 lists only four OTC pairs ($n = 4$). Provide post-hac power curves or minimum detectable effect sizes for key traits (SRL, AMF, P_o). This guards against Type-II error given low replication.

For all GLMMs, report marginal/conditional R^2 (Nakagawa 2017) and AIC delta to the next-best error structure in the Supplement.

2. P-budget closure

- The 78 % decline in soil P_o is large yet not accompanied by a measurable increase in P_i or microbial-P. Add a back-of-the-envelope mass balance: convert P_o loss ($\mu\text{g g}^{-1}$) to kg ha^{-1} for the 0-15 cm layer and compare with estimated extra P uptake needed to sustain the +67 % biomass growth reported in Damasceno et al. 2024. Discuss possible sinks (leaching, occlusion, hyphal translocation to litter layer).

3. Litter-layer P dynamics

Litter-bag P data are shown only as total P. Supply P_o and P_i concentrations in litter residues; this is critical to confirm that litter-root phosphatase activity actually mineralises P_o .

4. Model integration

The final paragraph correctly cautions about long-term P limitation. Strengthen by inserting one sentence that translates observed trait shifts into parameters used by terrestrial biosphere models (e.g., root exudation C-cost, AMF C-drain,

Langmuir P sorption). A table with suggested parameter values or % changes would be helpful.

5. Minor suggestions

Discuss whether 300-ppm enrichment (≈ 760 ppm) might overestimate responses expected under SSP5-8.5 2100 (≈ 550 ppm). A single sentence placing the study in the context of realistic 21st-century [CO₂] trajectories is sufficient.

Provide DOI or repository link for raw data and R scripts in the Data Availability statement.

Version 2:

Reviewer comments:

Reviewer #1

(Remarks to the Author)

I checked the revised manuscript. I feel the authors significantly improved the MS and I recommend to accept it now.

Reviewer #2

(Remarks to the Author)

I appreciate the authors' detailed rebuttal and the substantial additional analyses. All of my minor concerns—(i) Statistical transparency, (ii) P-budget closure, (iii) Litter-layer P dynamics, and (iv) Model integration—have been satisfactorily addressed. The manuscript now presents a balanced discussion and provides the data required to support its conclusions. I support publication with acceptance.

Amazonian understory forests change phosphorus acquisition strategies under elevated CO₂

REVIEWER COMMENTS

Reviewer #1 (Remarks to the Author):

The Ms by Martins et al., tried to explore the impact of eCO₂ on P acquisition strategies in Amazonian understory tree communities, addressing the role of root morphological changes and AMF in overcoming P limitation. Upon reading the relevant literature and current MS by Martins et al., I could say this study is timely and of high ecological relevance, particularly in the context of nutrient-limited tropical forests facing climate change. The experimental design using OTCs in P-depleted Amazonian soils is robust and innovative. However, the MS currently suffers from critical conceptual, structural, and mechanistic weaknesses, and the interpretation at times outpaces the data presented. Significant rewriting, clarification of causality, and broader ecosystem framing are required. Here are few of my comments/suggestions.

We sincerely thank the reviewer for the careful evaluation of our manuscript and for the constructive feedback provided. In the revised version, we have thoroughly addressed these points through substantial rewriting and restructuring of several sections. The detailed changes resulting from the review suggestions are outlined below.

- The experiment was carried out with minimum treatments and with 4 replicates. Given the small spatial and temporal scale of the OTC setup, how do the authors justify statements about Amazon-wide carbon sink resilience and ecosystem functioning? As reported in the abstract. “decisive to the Amazon forest’s resilience”.

We agree with the reviewer that our experimental design does not represent the entire Amazon forest. We have revised the sentence to ensure consistency with our results and to frame it appropriately within the scope of our study. The review can see some examples of the changes in the abstract and in the final paragraph of the manuscript, respectively:

Lines: 45-47:

“Such spatially divergent adjustments in nutrient acquisition strategies by roots may critically regulate plant–soil carbon and phosphorus coupling and should be incorporated into assessments of Amazon forest resilience under future climate change.”

Lines: 211-215:

“Our findings point to the potential for increasing P supply due to an increase in the P cycling in the Amazon forest, yet do not rule out progressive P limitation of plant productivity with eCO₂, as increasing P demand ultimately exceeds supply. Therefore, given Amazon’s central role in sustaining local and global climate and C balance, understanding whether these shifts can sustain long-term forest growth under rising CO₂ is critical.”

- There needs some more details about soil samples i.e., depth of soil sampling or soil profile characterization? This is important for nutrient and texture context, how often measurements of root traits, soil nutrients, or other variables were taken relative to the CO₂ treatment timeline?

R: Thank you for pointing this out. Our study focused on root and microbial dynamics in the litter layer on top of the soil, as well as in the upper 15 cm of soil. The soils are characterized as Geric Ferralsols, which are highly weathered and exhibit low total and available P levels. We have now added more details about the soil layer being investigated in our study.

In addition, more information about the physical and chemical soil characterization can be found in lines 221-223 and 262-266, respectively.

“The soil is clay-rich (67.7% clay, 19.9% sand, and 12.3% silt) and highly weathered Geric Ferralsols, with a pH of 3.94 and a low concentration of total phosphorus and rock-derived nutrients.”

“Soil chemical and physical properties inside the OTCs follow the same pattern previously described for the Ferralsols in Central Amazonian (Quesada et al, 2010), up 15 cm deep the soils present an average of 102.58 mg kg⁻¹ of total P, where 47% is P residual and 52% is extractable P, 3.7 of pH, 3.06% of carbon, and 0.24% of nitrogen (see details by OTC in Table S1).”

Regarding the frequency of measurement and data collection, we have added a new paragraph in the Methods section to describe in detail all the experiments and sampling procedures. Lines 242-255

“Sample collection occurred in two phases (Fig. S3). The first baseline phase (2019) encompassed all preparatory procedures, including OTC installation and an initial sampling campaign. We installed ingrowth cores in the litter layer and in the soil in February 2019 to study baseline root productivity and collected them in August 2019. Also in August 2019, we conducted a first soil sampling campaign to characterize soil nutrient concentrations, microbial biomass, and activity rates. Ingrowth cores were reinstalled in November 2019, immediately before starting the CO₂ enrichment. The eCO₂ phase started in November 2019. Ingrowth cores to investigate litter-based fine-root productivity were assessed after eight months (November 2019 to July 2020); soil-based fine root productivity was assessed by collecting ingrowth cores after twelve months (November 2019 to November 2020). Additionally, we collected soil samples in November 2020 after twelve months of eCO₂ to investigate root biomass, soil nutrient concentrations, microbial biomass, and enzyme activity. A litter-decomposition experiment was initiated in September 2020 and was concluded with a single harvest in September 2021. When available, baseline data were used to minimize a possible natural spatial variability of variables between OTC and seasonal effects.”

Reference:

Quesada, C. A. *et al.* Variations in chemical and physical properties of Amazon forest soils in relation to their genesis. *Biogeosciences* 7, 1515–1541 (2010).

Abstract

- Decisive to the Amazon forest's resilience..... these claims feel like stronger and more overestimated. Authors may choose soften speculative because there is lot to determine and analysis before going to final decisions.

R: We agree with the reviewer that the size of our study is not big enough to scale up to the entire Amazon forest. However, we believe that our study provides sound initial evidence that the spatially divergent responses of roots in the litter layer and in the soil are important and a crucial missing piece in understanding responses in larger-scale experiments. In particular, we want to emphasize that litter-layer roots must be considered in this ecosystem to fully understand its responses to elevated CO₂ and other climate change factors. In order to address the possible conclusions supported by our results, we changed the sentence, lines 43-47:

“Our findings suggest that eCO₂ intensifies the competition for P between plant roots and soil microorganisms, leading to changes in litter and soil P pools, exacerbating already strong nutrient constraints. Such spatially divergent adjustments in nutrient acquisition strategies by roots may critically regulate plant–soil carbon and phosphorus coupling and should be incorporated into assessments of Amazon forest resilience under future climate change.”

- Capacity to act as a future carbon sink..... this is well-known fact that the forest already acts as a carbon sink, authors should change words like future role or continued capacity.

R: We thank the reviewer for the suggestion and changed the sentence to: *“The potential for the Amazon forest to continue functioning as a carbon (C) sink strongly depends on soil nutrient availability, particularly phosphorus (P), and on plants' ability to adjust nutrient acquisition strategies.”* (lines 34-36).

- I don't know why the authors used citations in the abstract. References should be omitted from the abstract and there are some nonscientific/ inappropriate words, they should be replaced by appropriate words. Here are a few examples, spatial segregation, less productive.

R: The entire manuscript was restructured to follow the Nature Communications guidelines. More specifically, references were removed from the abstract, and non-scientific terms were replaced with more scientific terminology.

- Please provide abbreviation with arbuscular mycorrhizal fungi, as AMF etc and results should be written in the past tense.

R: We changed this as suggested by the reviewer.

Main text.. should be replaced by Introduction, please follow the journal guidelines.

R: We changed the structure, as suggested.

- Authors sometimes used phosphorous, and sometimes abbreviation P, please be consistent when using abbreviation and full name, please check throughout MS.

R: We checked throughout the manuscript to make consistent use of the terms and abbreviations.

- 57-59 It sounds like.....nutrient feedbacks and experimental data are not accounted for in Earth system models,” which is confusing. Please add subject of ...not accounted for..... How do current Earth system models implement or fail to implement nutrient feedback loops, especially P and N limitations?

R: Thank you for pointing this out. We have changed the sentence, see lines 53-57.

- L59 suggest,.....please remove comma

R: Removed as suggested.

- 69-71, very long sentence and too much general information. Please split it and concise information

R: We changed the sentence to have a clear message.

Line (63-66): “*In many highly weathered soils of the Amazon basin, a large fraction of P is not accessible and strongly adsorbed to the soil matrix (residual P), adsorbed to iron and aluminum oxides (mineral P), or bound in organic forms (Po), and only a relatively small inorganic P (Pi) fraction in the soil solution is accessible to plants and microbial communities (Quesada et al, 2010; Walker and Syers, 1976).*”

References:

Quesada, C. A. *et al.* Variations in chemical and physical properties of Amazon forest soils in relation to their genesis. *Biogeosciences* 7, 1515–1541 (2010).

Walker, T. W. & Syers, J. K. The fate of phosphorus during pedogenesis. *Geoderma* 15, 1–19 (1976).

- Should be ...microbial communities, rather than microbial community

R: We changed this, as suggested by the reviewers.

- different predominant P forms available?? What are the predominant P forms, please be specific.

R: Thank you for pointing this out. The predominant P form in the studied soil refers to organic vs. inorganic P fraction. We added a sentence with additional information about the soil P fractions and availability, lines 63–66:

- Some words are not defined earlier, for examples, trait syndromes, root economic space,, collaborative axis...

R: We thank the reviewer for this helpful comment. We have revised the manuscript to provide clear definitions of these key terms upon their first mention. In addition, we have added more context about root strategies depending on soil P availability. See for instance in lines 77-80.

- What is the relative impact of acid phosphatase activity vs. mycorrhizal uptake in Pi acquisition under elevated CO₂? What proportion of P uptake in Amazon understory comes from litter vs. mineral soil under eCO₂? Please provide explain for better understanding the background.

R: We thank the reviewer for this comment. We have expanded this section to provide more background and clarify the relative roles of acid phosphatase activity and mycorrhizal uptake in Pi acquisition. Acid phosphatases can hydrolyze phosphate (Pi) from organic phosphorus (P) compounds, making it available for plant uptake. In contrast, mycorrhizal associations facilitate Pi acquisition from both litter and mineral soil pools through extensive hyphal exploration, rather than contributing to Po mineralization.

We fully agree with the reviewer that it would be great to partition plant P uptake from litter vs. mineral soil, but this is not possible with our experimental setup. However, we can estimate net changes in Po and Pi pools in the litter layer and the soil and infer possible pathways of plant vs. microbial P utilization. However, we cannot quantify gross transfer rates of Pi from one pool to the other, nor split the proportion

of plant P uptake to the respective sources. Further, mechanistic studies, using radioisotope tracer applications are required to determine this information.

- L91 Does “it” mean to Pi or organic P?, please be specific

R: We revised the sentence to clarify the difference between the P forms. Specifically, at the beginning of the sentence, we refer to the organic P (Po) that will be hydrolyzed by acid phosphatase enzymes and become available to plant uptake as inorganic P (Pi).

The restructured sentence is presented in lines 73-75:

“In addition, they have been found to stimulate the mineralization of Po from decaying leaves and wood through phosphatase enzymes (Martins et al. 2021, 2023).”

- increased photosynthetic capacity but without changing aboveground biomass growth.... Please avoid repetition, as explained previously, I suggest to add possible mechanisms for belowground response or how hyper-diversity could change dynamics..

R: The text was checked to avoid repeating the information. Additionally, in the introduction, we added more information about the different root strategies and the plant-root interactions with the microbes to increase the P acquisition efficiency, and a specific hypothesis for our study was added, including more details about what we expect.

Lines 97-101: *“We hypothesized that under eCO₂, plants have used the extra C available to i) increase soil root biomass and productivity, ii) further invest in mining P from Po-rich sources like the organic litter layer, and iii) increase investment in AMF symbioses. In addition, we investigated the consequences of CO₂ fertilization on soil nutrient composition and microbial community dynamics to gain a more comprehensive understanding of the effects of eCO₂.”*

- Eucalyptus should be italicized and provide full scientific name if authors are referencing plant, *Eucalyptus globulus*

R: . We have corrected the text to italicize *Eucalyptus*. We have double-checked the study, and it does not specify the Eucalyptus species (Ellsworth et al, 2016). We maintained the approach as described in the reference paper.

References:

Ellsworth, D. S. *et al.* Elevated CO₂ does not increase eucalypt forest productivity on a low-phosphorus soil. *Nat. Clim. Chang.* 7, 279–282 (2017).

- at the study site.... Please add the site name.

R: We changed the sentence to specify the name of the site as suggested by the review (line 103 in the revised manuscript).

- Strikingly.. please avoid non-scientific words

R: The term “strikingly” has been removed to maintain a strictly scientific tone.

- 114, were set up in pair ... here authors meant four pairs???

R: We have clarified the sentence to specify that four pairs of open top chambers were established. The revised text now reads:

“A total of eight OTCs (with a 2.4 m diameter and 3 m of height) were established in four pairs, distributed across a plateau within a primary forest.” (lines 104–105 in the revised manuscript).

• we aim.. should be we aimed to...

R: We changed it as suggested.

• whether the authors randomize the chamber placement or account for spatial gradients light, slope etc. please also mention this.

R: We thank the reviewer for this comment. We confirm that chamber placement was randomized to minimize potential spatial bias. The four pairs of open-top chambers (OTCs) were randomly distributed across a relatively homogeneous plateau area. We have now clarified this in the Methods section of the revised manuscript.

• How representative are these understory tree species of the broader Amazon understory in terms of function and diversity?

R: The understory plant community evaluated in our study represents a naturally regenerating assemblage of multiple species typical of the Central Amazon *terra firme* forest encompassing typical plant functional types for Amazonian understories, including shade-tolerant and intermediate light-demanding species with contrasting leaf, stem, and growth traits, shaped by light limitation, nutrient scarcity, and microsite variability (Damasceno et al 2024). Although our experiment does not capture the full taxonomic diversity of the Amazon basin, it provides a representative snapshot of the functional spectrum and ecological strategies that dominate these forests. Importantly, our experimental design focused on community-level mechanisms, specifically whether eCO₂ causes modifications to plant nutrient acquisition strategies at the root level in the soil and litter layer, as well as changes in litter and soil nutrient dynamics and microbial activity. Therefore, we believe that the results are broadly applicable to understanding the functional dynamics of tropical understory trees in the Central Amazon.

References:

Damasceno, A. R. *et al.* In situ short-term responses of Amazonian understory plants to elevated CO₂. *Plant Cell Environ.* 1–12 (2024).

• ,,,,under eCO₂, On the one hand.. . please remove comma after eCO₂

R: We changed it as suggested.

• Authors used two different terminologies, soil-based roots, roots in the soil, please be consistent with the use of terminologies.

R: We changed throughout the manuscript and unified the terminology to soil-based roots and litter-based roots.

• a trade-off between root production and symbiosis..... what does this trade-off suggest.....?

R: We could not identify the exact text section the reviewer was referring to. In general, we considered trade-offs that one function is being over-expressed as an adaptation e.g., to low nutrient availability at the cost of a reduction of another function. For instance, in systems with high root productivity and turnover, mycorrhizal colonization could be reduced potentially due to the short-lived nature of fine roots.

- arbuscular mycorrhizal (AMF) colonization.... should be arbuscular mycorrhizal fungi (AMF).....please use abbreviations once full name is used. Please check throughout MS.

R: We have made the changes as suggested by the reviewer. In addition, we carefully reviewed the entire manuscript to ensure consistency and standardization of all abbreviations.

- roots are favoring..... roots don't favor strategies,..... plants do. Please change

R: We appreciate the reviewer's observation. We changed the phrasing to make it clear.

- once you used full name, root phosphatase activity, please abbreviate next.....

R: We changed it as suggested.

- 148 In summary, in litter-based roots....please avoid concluding after every paragraph.

R: We reduced providing conclusions, but kept them in sections, where we considered them to fit well.

- Please use abbreviations for phosphorus.

R: We changed it as suggested by the reviewer.

- A 189% reduction is not mathematically logical unless it states more than complete loss. Perhaps meant reduction to 11% (89% decrease) or activity decreased by 89%. Please double check the data.....

R: Indeed, this is an important observation. We have checked the data, and this was calculated as the percentage decrease between the two averages of the Cellobiosidase enzyme (i.e., aCO₂ and eCO₂ averages). We use the following equation to calculate the percentage decrease: $((eCO_2 - aCO_2) / aCO_2) * 100$, and we confirm that the percentage decrease of 189% is possible. However, to avoid confusion, we removed the percentage of decrease from the main text.

- You make several causal claims (e.g., eCO₂ leads to enhanced P mobilization) that are well-supported I suggest to explicitly framed as interpretations, not facts.

R: We screened the text and tried to indicate more clearly where we suggest or conclude from our data rather than stating results as facts.

- 174-178-- last sentence is quite long and multiple arguments at once,,,, please avoid conclusion after every paragraph.

R: Thank you for pointing this out. We have now changed the structure of the results to better adjust the discussion section.

- 170-172 enzyme allocation toward Pit should be functional P limitation

R: We did not understand this suggestion. This section has been changed to improve the understanding.

- Is it possible that microbial community composition changed (even if microbial P pool didn't)?

R: We thank the reviewer for this insightful comment. Although we did not assess microbial community composition, we indeed acknowledge that shifts in microbial taxa or functional groups could occur under elevated CO₂, even in the absence of detectable changes in the total microbial P pool.

- Please use abbreviations AMF, P, eCO₂ etc please be consistent with the use of elevated eCO₂ or enhanced eCO₂

R: We have carefully revised the manuscript to standardize all abbreviations and to ensure consistent use of the term referring to elevated atmospheric CO₂ concentrations.

- progressive P limitation.... What is the cause of it and please also add explanation.

R: We agree that further clarification is needed. We have considered progressive P limitation analogous to progressive N limitation observed in temperate FACE experiments (see, for instance, this paper, Norby et al., 2010). We here consider progressive P limitation to the gradual decline in P availability in the soil relative to increasing plant and microbial demand under elevated CO₂ conditions needed to facilitate a CO₂ fertilization effect – i.e., stimulated growth and productivity, and thereby could also increase C inputs to the soil, but also stimulate plant nutrient uptake and microbial activity. Since P derives from bedrock weathering, and is a limited resource, and in the studied ecosystem is already particularly low, the replenishment of P_i from mineral weathering and P mineralization is low. In particular, with increasing CO₂, P can therefore become increasingly limiting and progressively limit an initial fertilization effect.

References:

Norby, R. J., Warren, J. M., Iversen, C. M., Medlyn, B. E. & McMurtrie, R. E. CO₂ enhancement of forest productivity constrained by limited nitrogen availability. *Proceedings of the National Academy of Sciences* 107, 19368–19373 (2010).

- future research should evaluate whether these root and microbial adjustments.... authors provided too many general future directions that is somehow is not possible, and no practical use, I suggest to provide simple ones, and practical, and feasible.

R: This is a good point. Considering that we have few data about the belowground dynamics in the Amazon forest, and specifically about the elevated CO₂ effect on this process, we believe that many different studies need to be carried out to increase our knowledge. In our manuscript we aimed to point towards information and data gaps, how often overlooked and understudied to root and soil interactions can help to gain a more holistic picture of tropical forest functioning and to understand ecosystem level responses to elevated CO₂ and potentially other climate change factors.

- References have lot of issues,

R: We double-checked the references and their formatting.

- Authors sometimes used et al., and sometimes used all the authors name, please be consistent

R: We have now edited and adjusted to the style of Nature Communications.

- Don't need to capitalized first letter of the title of the paper in the reference

R: We are not sure what the reviewer is referring to.

- In, CO₂² should be subscript.

R: Thanks for spotting this. We have changed it accordingly.

- Authors used lot of references, even cited for general statement, please reduced number of references.

R: We screened the references and adjusted accordingly.

- Tweedie distribution family, it should be Tweedy seem like typo

R: Thanks for spotting this. This was changed accordingly.

- Some sentences are quite long and complex. Breaking them into shorter sentences

R. We have shortened several sentences as suggested.

- Use past tense consistently for methods already performed.

R: We change as suggested.

- When introducing abbreviations i.e., AMF for arbuscular mycorrhizal fungi), define once and use thereafter.

R: We checked through the manuscript to use the abbreviations after the first definition.

I suggest to add a map of the study area, so that it would be clear how many points, locations and area was covered on the actual map.

R: We appreciate the suggestion. We have now added a study area map in the supplementary material that presents the number and distribution of the Open Top Chambers across the experimental study area.

In Table, S2, don't need to italic some of the words. i.e, Baseline Elevated CO₂ experiment Delta CO₂ effect in Table S4, please subscript of the powers of the chemical names, also check throughout Ms... NaHC₃... NaHCO₃

R: We thank the reviewer for pointing this out. We made the changes as suggested and also checked the manuscript throughout to standardize the chemical nomenclature.

There is contradiction between main text and Figure S1. 3 m height and 2.4 cm diameter, 2.4 cm is quite small diameter, please double check.

R: We appreciate the observation. The right size of the OTCs is 2.4 m in diameter and 3 m in height. We changed the information in Figure S1 to be consistent with the main text.

Figure S3, The data represent the period between January 2020 and October 2021, samples was done just after peak of Covid-19, there were some challenges in taking readings, climate factors, etc

R: This is a very important comment, and I appreciate the opportunity to clarify our sampling timeline, the limitations, but also the success of the experiment conducted under very difficult circumstances. In fact, our experiment timeline is aligned with COVID-19 pandemic, and the initially planned timeline was impacted by lockdown-related regulations. However, with a very supportive team and impressive logistical support, it was possible to adapt the sampling for some periods, ensuring the maximum safety for the team. The AmazonFACE project's logistical support had access to the experimental site every week over the entire period (November 2019 - October 2021) to ensure that the elevated CO₂ system was working properly (this data can be verified in Figure S4).

In addition, the baseline sampling campaign conducted in 2019 provided us with very powerful data about the spatial heterogeneity of litter, root and soil parameters within the OTCs before starting the elevated CO₂ treatment.

Using this data gives us more confidence in an elevated CO₂ effect because it accounts for pre-existing variability.

Figure S3, some texts are blurred, difficult to read. please increase Quality

R: We increase the quality of the figure to guarantee a better visualization of all the details.

Reviewer #2 (Remarks to the Author):

This manuscript provides timely experimental evidence on how Amazonian understory plants adapt root P-acquisition strategies under elevated CO₂. The in situ OTC experiment in P-depleted soils addresses

a critical gap in understanding tropical forest resilience to climate change. The dual-strategy mechanism (litter "DIY" vs. soil "outsourcing") is compelling and advances ecological theory. While the findings are well-supported, however, I have a few comments regarding some methodological limitations and mechanistic ambiguities to enhance logical flow and clarity.

R: We thank the reviewer for their thoughtful and encouraging comments. We acknowledge the reviewer's constructive remarks regarding methodological limitations and mechanistic ambiguities. We have carefully addressed these points in the revised manuscript to improve clarity and logical flow. Specific responses to each comment are provided below, along with corresponding revisions to the text.

1. Statistical Power and Replication

With only $n=4$ OTCs per treatment, statistical power is limited despite GLMMs accounting for spatial variability.

R: We are aware of the limitations of having only four OTCs per treatment in such an ecological complex ecosystem. However, given the logistical complexity of longer-term in situ experiments in tropical forests, such as the Amazon forest, this replication level aligns with established standards in similar studies. Most experiments testing elevated CO₂ concentrations that have been conducted worldwide have a replication number of three due to cost and logistical considerations. See below some of the examples:

Oren, Ram, et al. "Soil fertility limits carbon sequestration by forest ecosystems in a CO₂-enriched atmosphere." *Nature* 411.6836 (2001): 469-472.

Norby, Richard J., et al. "Forest response to elevated CO₂ is conserved across a broad range of productivity." *Proceedings of the National Academy of Sciences* 102.50 (2005): 18052-18056.

Ellsworth, David S., et al. "Elevated CO₂ does not increase eucalypt forest productivity on a low-phosphorus soil." *Nature Climate Change* 7.4 (2017): 279-282.

2. Mechanism of Soil Organic P Decline

The 78% reduction in soil organic P (Fig. 1c) is mechanistically ambiguous:

Not linked to root/microbial APase activity (Fig. S8/S19b).

Attribution to "reduced organic P input from litter" (L162) contradicts observations: Litter-layer roots mobilize P, which should increase soil P inputs.

R: We appreciate the reviewer's insightful comment on the mechanism underlying the decline in soil organic P. We agree that the observed reduction in soil Po cannot be explained by root or soil microbial APase activity. In an earlier study, we found evidence that litter Po inputs are linked to soil Po inputs (Schaap et al 2021), but also that fine roots play an important role in mobilizing Po to Pi and intercept the Pi from the litter layer (Martins 2021). This confirms that the root mats observed in the litter layer of some regions in the Amazon forest are highly efficient in acquiring and taking up newly available nutrients before they can be leached into the soil (Cuevas and Medina 1988). Hence, we think that the argument that the reduced organic P (Po) input from the litter layer does not contradict the decrease in soil P, but rather points out that with increasing demand for limiting nutrients, plants may even further intensify nutrient recycling from the litter layer.

We revised the manuscript to better explain the links between litter and soil Po and Pi cycling under eCO₂, as explained above, the P mobilization process in the litter layer can impact the soil P dynamics, and added further potential explanations for the observed in the lines 147-155:

Although P_o is the largest P fraction in Ferralsols/Oxisol soils (Quesada et al 2010), it becomes available only after mineralization through microbe or plant root released phosphatase-catalyzed hydrolysis to P_i (Wasner et al., 2023). However, as in our study, we did not detect changes in root and microbial APase activity (Fig. S9 and Fig. S18d), the decrease in P_o could be attributable to a change in exudates released into the soil. For instance, enhanced exudation of organic/carboxylic acids could mobilize not only inorganic but also organic compounds from the mineral soil matrix, which could concomitantly liberate P_o and P_i (Keiluweit 2015). Similarly, photosynthates distributed by roots or AMF hyphae could foster phosphate-solubilizing bacteria in the rhizosphere or hyphosphere⁴³ (Zhang et al., 2016). Moreover, the decrease in P_o could have been caused by other plants or microbial phosphatases or phytases that were not determined in our study (Turner et al, 2008).

and a new link for the P dynamic in the litter layer can be found in lines 170-177:

In addition to the observed depletion in soil P_o under eCO_2 , one year later, we also observed a significant decrease in total P concentration in the decomposing leaf litter, without any change in litter C decomposition (Fig. S10). This suggests that the adaptations of litter-based fine roots to eCO_2 may have longer-term consequences for litter P recycling, with subsequent effects on soil P cycling. Despite the different collection times making it difficult to directly link the P dynamics in the soil and litter layer, our results strongly suggest that the plant community's strategy of investing in multiple root nutrient acquisition strategies impacts the P dynamics in the soil and litter layer, with consequences for the plant root and microbial competition.

In addition, more details about the timeline of the sample collections were included in the method section, lines 242 to 255.

References:

- Schaap, K. J. et al. Litter inputs and phosphatase activity affect the temporal variability of organic phosphorus in a tropical forest soil in the Central Amazon. *Plant Soil* 469, 423–441 (2021).
- Martins, N. P. et al. Fine roots stimulate nutrient release during early stages of leaf litter decomposition in a Central Amazon rainforest. *Plant Soil* 469, 287–303 (2021).
- Cuevas, E. & Medina, E. Nutrient dynamics within amazonian forests - II. Fine root growth, nutrient availability and leaf litter decomposition. *Oecologia* 76, 222–235 (1988).
- Quesada, C. A. et al. Variations in chemical and physical properties of Amazon forest soils in relation to their genesis. *Biogeosciences* 7, 1515–1541 (2010).
- Wasner, D. et al. Tracing 33 P-labelled organic phosphorus compounds in two soils : New insights into decomposition dynamics and direct use by microbes. *Frontiers in Soil Science* 3, (2023).
- Keiluweit, M. et al. Mineral protection of soil carbon counteracted by root exudates. *Nat Clim Chang* 5, 588–595 (2015).
- Zhang, L. et al. Carbon and phosphorus exchange may enable cooperation between an arbuscular mycorrhizal fungus and a phosphate-solubilizing bacterium. *New Phytologist* 210, 1022–1032 (2016).

Turner, B. L. Resource partitioning for soil phosphorus: A hypothesis. *Journal of Ecology* 96, 698–702 (2008).

Recommendation:

Test correlations between litter-root traits (SRL, diameter) and soil P decline.

R: We appreciate this suggestion and have also been considering this option. However, due to the different timelines of the sample collections, we believe that it would not be sound to correlate the variables with each other. We carefully reviewed the manuscript and made numerous changes to the results and discussion to clarify our sampling timeline and the connections between the processes in the litter and soil layers. We are confident that we have a clear description of our results and a strong hypothesis underlying the processes that explain the findings.

Discuss alternative drivers (e.g., microbial priming, unmeasured phosphatases).

R: We acknowledge that both other processes or other phosphatases not analyzed in this study may also contribute to the observed decline and could play a role in the decrease in soil P_o . In the revised manuscript, we expand the discussion on alternative explanations for the organic P decrease under eCO_2 (lines 149-155).

3. Microbial Competition Narrative

Reduced C-degrading enzymes (CB, BG) suggest microbial P limitation (L167–172), yet:

Microbial biomass P (P_m) did not change (Fig. S17c).

AMF colonization increased (outsourcing strategy), implying reduced plant-microbe competition.

R: We appreciate the reviewer's thoughtful observation. As discussed in lines 159–164, the decline in extracellular enzyme C:P ratio under elevated CO_2 , was particularly driven by a decrease in the CB and BG activity, which suggests a reduction in the demand of enzymes needed to degrade more complex C containing compounds, and could indicate that the soil microbial communities may satisfy their C-demand by potentially increased labile C exudation (Reay et al., 2025). Simultaneously, the lack of change in phosphatase production indicates either a maintained or increased microbial P demand. Our data showed that microbial biomass P and soil P_i levels remained unchanged. Hence, we can conclude that the microbial community may not have been affected yet by the changes in soil P_o . Together with the observed increase in root AMF colonization, our results point rather towards enhanced plant P uptake efficiency. However, we found no evidence in the literature that increased AMF colonization reduces plant-microbe competition. Therefore, our results suggest that both plants and microbes are adjusting their strategies to mobilize P, likely intensifying plant-microbe competition for phosphorus.

Reference:

Reay, M. K. *et al.* Elevated CO_2 alters relative belowground carbon investment for nutrient acquisition in a mature temperate forest. *Proceedings of the National Academy of Sciences* 122, (2025).

4. Transient vs. Sustained P Mobilization

Litter-layer P mining may be short-term effective but unsustainable:

Litter P is finite and input-constrained (L211–212).

R: As discussed in lines 179-185, our results suggest that the modulation of multiple root nutrient acquisition strategies in the litter layer and soil may enhance P acquisition efficiency in the short term. However, because eCO₂ can increase both plant and microbial P demand while P availability remains dependent on internal recycling, this may lead over time to progressive P limitation. Our results highlight the importance of long-term eCO₂ experiments to capture shifts in processes and mechanisms over time, offering deeper insight into belowground dynamics in Amazonian forests.

Some minor comments:

Lines 428–429: Root Productivity Metrics: Unify terminology (e.g., "mg cm⁻² day⁻¹" vs. "mg year⁻¹"; Methods P14–15).

R: Thank you. The different terms reflect different units of measurement: root productivity was initially expressed in mg cm⁻² day⁻¹ and then scaled to mg year⁻¹. We have revised the sentence to clarify this distinction. New manuscript version, lines 296 - 300.

Line143: Phosphatase Activity: Resolve discrepancy: Fig. 1a lists "APase" with p-values, but text states no change.

R: Thank you. We have double-checked all figures to resolve any discrepancies.

Line154: Litter Decomposition: Discuss why mass loss was unaffected despite P decline.

R: We thank the reviewer for this comment. The P mineralization can be decoupled from C decomposition, meaning that phosphatase enzymes can hydrolyze organic P independently of changes in litter mass loss (Martins et al., 2021). We added more details about this in the result-discussion section (lines 170-177).

Reference:

Martins, N. P. *et al.* Fine roots stimulate nutrient release during early stages of leaf litter decomposition in a Central Amazon rainforest. *Plant Soil* 469, 287–303 (2021).

Line 126: AMF Colonization: Report absolute % colonization (not just % change;).

R: We add the information as suggested. In the new manuscript version, line 144.

Figures & Tables

Fig. 1: Specify error bars (95% CI or SE? L335); include n=4 in caption.

R: We revised the figure caption and made clear the information raised by the reviewer.

Extended Data: Add schematics of OTC/trench design (Fig. S1) to Methods.

R: We appreciate the suggestion but decided to keep it as Figure S1 in the supplementary material because it represents rather general information about the manuscript. However, we provide a comprehensive description of the OTCs in the main text (Lines 227-241).

Additional Suggestions:

Supplemental Analyses: Correlate root trait shifts with soil P fractions.

R: As we already explained earlier, due to the different timelines of the sample collections, we believe that it is not justified to correlate root trait with soil P fraction shifts.

Give a figure about P cycling or P budget in different layers among soil, litter and trees.

R: We thank the reviewer for the suggestion. Nevertheless, we do not have the data for all plant compartments necessary to construct a complete P cycle diagram. In our manuscript, we focus on understanding the different plant root P acquisition strategies that interact with microbial activity, which could influence P cycling, rather than understanding P cycling in detail from an ecosystem perspective. We present an illustration integrating the main effects of elevated CO₂ on root nutrient acquisition strategies in both the litter layer and the soil and impacts on soil P dynamics (Figure 5).

Table S1 should describe proportion of AMF and ECM trees.

R: We appreciate the suggestion. The majority of trees in tropical forests are AMF, and only very few reports are given for EMF trees; hence, they may play a rather negligible role in this ecosystem. Moreover, during sampling, we found no visual evidence for EMF structures. We have checked databases for potential AMF vs. EMF trees within the species in the OTCs and also found no reports so far for potential EMF trees.

Future focus: What is the mechanism of mycorrhizal plants allocating C within their roots and mycorrhizae? How to accurately distinguish the contribution of the direct pathway via roots and the mycorrhizal pathway via AM fungal hyphae to plant P acquisition?

If there is a shift in inputs to the soil (like root biomass or hyphal/mycelium inputs), this could change the quality/quantity of C in the soil and C residence time. Similarly, as suggested by reviewer 1, to understand the origin of and pathway of P uptake into plants, it would need either targeted partitioning experiments or radioisotope tracer experiments, which are difficult to do under field conditions, or they would need pot experiments, and might be very interesting follow-up studies, but were beyond the scope of this study.

Amazonian understory forests change phosphorus acquisition strategies under elevated CO₂

REVIEWER COMMENTS

Reviewer #1 (Remarks to the Author):

This MS provides robust experimental evidence that elevated CO₂ can reshape phosphorus acquisition strategies in Amazonian understory forests through coordinated changes in root traits, AMF, and soil P cycling. The work is original, methodologically sound, and of broad relevance to understanding tropical forest resilience under climate change. I reviewed the revised Ms and also read other reviewer comments, I found that authors revised the Ms and address all of the comments. Now quality of Ms is significantly improved. I therefore recommend acceptance of this Ms.

Comment; In the abstract, please use the subscript function in the MS word software for CO₂, check throughout Ms.

We sincerely thank the reviewer for the feedback and for recommending our manuscript for publication in Nature Communications. Additionally, we checked all subscript symbols throughout the manuscript to ensure they were formatted correctly.

Reviewer #2 (Remarks to the Author):

I appreciate the authors' detailed rebuttal and the substantial additional analyses. All of my major concerns—(i) statistical power justification, (ii) mechanistic explanation for the 78 % soil-P_o decline, (iii) clarification of the “microbial competition” narrative, and (iv) acknowledgement of temporal constraints on litter-P mining—have been satisfactorily addressed. The manuscript now presents a balanced discussion and provides the data required to support its conclusions. I support publication after minor revision.

We appreciate the reviewer's comments and support for publication in Nature Communications after minor revisions. We have addressed all the comments in detail below.

Essential revisions (no new experiments required)

1. Statistical transparency

Table S1 lists only four OTC pairs (n = 4). Provide post-hac power curves or minimum detectable effect sizes for key traits (SRL, AMF, Po). This guards against Type-II error given low replication.

Thank you for your comments and suggestions. In fact, our experimental design has a lower replication size, but given the logistical challenges of elevated CO₂ experiments, this is the standard replication number across different experiments worldwide. As suggested, we calculated the minimum detectable effect size for the key traits, and it's presented in the table below.

Table 1. **Example of the calculation of minimum detectable effect sizes for some key traits.** Data was analyzed using generalized linear mixed models; the treatment effect was included as a fixed factor, while spatial variability was accounted for by incorporating open-top chamber OTC pairs as a random factor. The delta CO₂ effect was calculated as the difference between post-enrichment collections and baseline measurements for both treatments.

	Delta CO ₂ effect						
	Est.	Std. error	Z	P	R ² m	R ² c	MDE – 80%
SRL (m g ⁻¹)	45.07	20.303	2.22	0.02	0.49	0.48	74.83
AMF (%)	30.25	9.65	3.13	0.01	0.43	0.43	53.77
Organic P (mg kg ⁻¹)	-7.72	2.59	-2.97	0.002	0.55	0.55	11.06

The variables shown in the table are the specific root length (SRL) in the litter layer, the arbuscular mycorrhizal fungi (AMF) in the soil, and the total organic P soil. We use the glmmTMB package to run the statistical models, and the model parameters are presented for each variable. Where: Est. = model estimate; Std. error = standard error; Z = wald statistic; P = p-value; R²m = variance explained by fixed effects; R²c = variance explained by fixed + random effects. MDE: minimum detectable effect sizes at 80%, calculated using the statistical model result with the power.t.test function.

When we interpret the MDE results in Table 1, we see that for the three variables tested, the MDE at 80% power exceeds the estimated value for our eCO₂ treatment. For our understanding, the MDE represents the data interval between our eCO₂ and aCO₂ treatments, in which eCO₂ shows a significant effect in 80% of the possibilities. Thus, effects smaller than the estimated MDE can still be detected, but with lower statistical power. Furthermore, the significant results presented here and throughout the manuscript do not contradict the MDE results. Rather, they indicate that our experimental design can detect moderate treatment effects, albeit with greater uncertainty. Our findings support a directional response of SRL, AMF, and soil organic P to elevated CO₂, while we explicitly acknowledge in the main text that the precision of the estimated effect sizes is limited by replication.

For all GLMMs, report marginal/conditional R² (Nakagawa 2017) and AIC delta to the next-best error structure in the Supplement.

Thank you for the suggestion. We have updated all the supplementary tables to report the marginal/conditional R² (Table S2-S5). However, we decided not to implement the “AIC information of the delta to the next-best error structure” as for most of the statistical models we used the ‘Gaussian family’ and evaluated assumptions of normality, homoscedasticity, and independence using the simulateResiduals function from the DHARMA package. We made a small change to the main text to be clearer about our approach (lines: 420-421):

“We fitted the model using the Gaussian statistical family as the basis for all variables, and when model diagnostics indicated inadequate fit, we tested other families (e.g., Tweedy and gamma). The assumptions for normality and homogeneity of the residuals using the simulateResiduals function from the DHARMA package.”

2. P-budget closure

- The 78 % decline in soil Po is large yet not accompanied by a measurable increase in Pi or microbial-P. Add a back-of-the-envelope mass balance: convert Po loss (μg g⁻¹) to kg ha⁻¹ for the 0-15 cm layer and compare with estimated extra P uptake needed to sustain the +67 % biomass growth reported in Damasceno et al. 2024. Discuss possible sinks (leaching, occlusion, hyphal translocation to litter layer).

Thank you for this great suggestion. We did some back-of-the-envelope calculations, which, however, required some assumptions and simplifications.

At our site, we found that eCO₂ led to a 65% increase in stem increment. For simplification, we used an average value of understory forest fraction of around 14% (Kunert and Aparecido, 2024) to estimate understory forest BM (around 49 Mg BM ha⁻¹) and C stocks (using an average of 47%), leading to values of around 23 Mg C ha⁻¹ in understory, or 2.3 kg C m⁻². Based on this stock, we calculated the average C increase of 65% in aboveground BM (a plus of approx. 1.5 kg C m⁻²). We similarly converted the BM to increased P needs for BM (assuming a (rather high) woody BM P concentration of around 124 mg P kg⁻¹, Dalling et al 2024. This resulted in an estimate of 0.6 kg P stored in BM m⁻², and an extra P demand for wood biomass production of 0.39 kg P m⁻². This does not include extra P for changes in leaf or root BM production, and therefore, the P budget is not complete. However, if we compare the increased demand with the 78% decrease in soil Po levels, representing approximately 2.2 kg P m⁻², this shows that only a fraction of Po lost may have been used by plants for BM production. Po could have become directly or, after enzymatic hydrolysis, occluded in the residual P pool and become unavailable; roots and hyphae may have re-allocated P into different soil layers.

However, as we do not have data on the P pools plant, we do not have a very complete P budget and would like to be rather careful with the numbers and would like not to over-interpret them. We therefore only stated a brief sentence in the main manuscript:

Line 169- 173: *'However, likely the Po lost exceeds the increased P demand by plants for the increased production (based on low P concentrations found in tropical forest trees Dalling et al., 2024), a fraction of Po could have also become directly or after enzymatic hydrolysis occluded in the residual P pool and become unavailable, or roots and hyphae may have re-allocated P into different soil layers.'*

Reference:

Kunert N. and Aparecido L., Ecosystem carbon fluxes are tree size-dependent in an Amazonian old-growth forest; *Agricultural and Forest Meteorology* 346 (2024) <https://doi.org/10.1016/j.agrformet.2024.109895>

Dalling *et al.*, (2024), Wood nutrients: Underexplored traits with functional and biogeochemical consequences *New Phytologist* (2024) 244: 1694–1708 doi: 10.1111/nph.20193

3. Litter-layer P dynamics

Litter-bag P data are shown only as total P. Supply Po and Pi concentrations in litter residues; this is critical to confirm that litter-root phosphatase activity actually mineralises Po.

We thank the reviewer for the comment. We agree that to identify the fractions of Po and Pi, it is important to confirm the Po mineralization. Although this data was not available for the litter layer in this study, in a previous study focused on detecting the role of roots in leaf litter decomposition, we identified root-excreted phosphatases as important accelerators of litter P loss (Martins et al., 2021). In this current experiment, we did not explicitly separate root from root-free litter decomposition and did not separate litter Pi and Po dynamics. Therefore, we just suggest that the decrease in total P could be due to root phosphatase mineralization.

Reference:

Martins, N. P. *et al.* Fine roots stimulate nutrient release during early stages of leaf litter decomposition in a Central Amazon rainforest. *Plant Soil* 469, 287–303 (2021).

4. Model integration

The final paragraph correctly cautions about long-term P limitation. Strengthen by inserting one sentence that translates observed trait shifts into parameters used by terrestrial biosphere models (e.g., root exudation C-cost, AMF C-drain, Langmuir P sorption). A table with suggested parameter values or % changes would be helpful.

Thank you for the suggestion. We have now added an additional table (Table S6) to the supplementary information with a summary of the effects induced by eCO₂.

5. Minor suggestions

Discuss whether 300-ppm enrichment (≈ 760 ppm) might overestimate responses expected under SSP5-8.5 2100 (≈ 550 ppm). A single sentence placing the study in the context of realistic 21st-century [CO₂] trajectories is sufficient.

We appreciate this suggestion and agree that the experimental setup may exceed SSP5 8.5 2100 expectations. However, given its complexity (higher CO₂ concentrations in forest understories, uncertainties of trajectories, interactions with other climate change factors), we felt a single sentence would not do it justice, so we have opted not to include it in this iteration.

Provide DOI or repository link for raw data and R scripts in the Data Availability statement.

The datasets generated and analyzed during the current study will be available in the Zenodo repository at <https://doi.org/10.5281/zenodo.18495501> after publication.